# scBridge embraces cell heterogeneity in single-cell RNA-seq and ATAC-seq data integration

Yunfan Li [1,5], Dan Zhang [2,5], Mouxing Yang [1], Dezhong Peng [1], Jun Yu[3], Yu Liu[4], Jiancheng Lv[1], Lu Chen [2] & Xi Peng [1] ✉

Single-cell multi-omics data integration aims to reduce the omics difference while keeping the cell type difference. However, it is daunting to model and distinguish the two differences due to cell heterogeneity. Namely, even cells of the same omics and type would have various features, making the two differences less significant. In this work, we reveal that instead of being an interference, cell heterogeneity could be exploited to improve data integration. Specifically, we observe that the omics difference varies in cells, and cells with smaller omics differences are easier to be integrated. Hence, unlike most existing works that homogeneously treat and integrate all cells, we propose a multi-omics data integration method (dubbed scBridge) that integrates cells in a heterogeneous manner. In brief, scBridge iterates between i) identifying reliable scATAC-seq cells that have smaller omics differences, and ii) integrating reliable scATAC-seq cells with scRNA-seq data to narrow the omics gap, thus benefiting the integration for the rest cells. Extensive experiments on seven multi-omics datasets demonstrate the superiority of scBridge compared with six representative baselines.

Single-cell RNA sequencing (scRNA-seq)[1] has been widely used and made great progress in the fields of biology and medicine. Recently, the advances in single-cell technologies have enabled profiling single cells from different layers, such as chromatin accessibility (scATAC-seq)[2,3], spatial transcriptome (Stereo-seq)[4], and proteome (ScoPE-MS)[5]. Integrating diverse omics data provides a chance to reconstruct a comprehensive molecular regulation network, and promote the development of precision medicine. In particular, scATAC-seq studies the physical structure of the genome by identifying open chromatin regions, while the dynamic remodeling of chromatin structure is one of the main mechanisms that affect transcription. Thus, the integration of scATAC-seq and scRNA-seq allows not only observing the differences at the transcriptional level but also understanding the reasons behind the differences from an epigenetic perspective[6].

Several methods have been proposed to integrate transcriptomic data[7–14], which could also be used for multi-omics data integration. However, suboptimal results would be achieved by directly applying those transcriptomic-oriented methods to integrate scRNA-seq and scATAC-seq, because the data distribution and sparsity level are vastly different across omics[15]. To address this issue, some efforts have been devoted to multi-omics data integration. Specifically, with the cross-omics pairing information, scAI[16] and MOFA+[17] could perform joint integration and clustering on multi-omics data. However, it is daunting to obtain such pairing information due to the prohibitive cost of multi-omics sequencing techniques[18,19]. As a remedy, the focus of the community has shifted to the scenario wherein the scRNA-seq and scATAC-seq data are sequenced independently, *i.e.*, unpaired. To integrate unpaired multi-omics data, the most common paradigm is first

[1]School of Computer Science, Sichuan University, Chengdu, Sichuan, China. [2]Key Laboratory of Birth Defects and Related Diseases of Women and Children of MOE, Department of Laboratory Medicine, State Key Laboratory of Biotherapy, West China Second University Hospital, Sichuan University, Chengdu, China. [3]School of Computer Science, Hangzhou Dianzi University, Hangzhou, Zhejiang, China. [4]School of Electronic and Information Engineering, Naval Aviation University, Yantai, Shandong, China. [5]These authors contributed equally: Yunfan Li, Dan Zhang. ✉e-mail: pengx.gm@gmail.com

independently conducting feature extraction in each omics, and then reducing the omics difference in features through manifold alignment[20,21], mutual nearest neighbor (MNN) correction[22], graph linking[23,24], and adversarial training[25]. A more straightforward solution is explicitly modeling the omics difference as a factor in matrix factorization[26,27]. Recently, considering the abundant annotated scRNA-seq data, scJoint[15] proposes to integrate multi-omics data under the semi-supervised learning paradigm.

Although various methods have been developed and achieved remarkable progress, most of them overlook the role of cell heterogeneity, not to mention exploiting it in the integration. To be specific, the objective of multi-omics integration is to reduce the omics difference while keeping the cell-type difference. However, due to the cell heterogeneity, even the cells of the same omics and type would have non-negligible variances which would make omics and cell-type differences less significant. As a result, it is daunting to model and distinguish the two differences, leading to suboptimal integration results. On the one hand, when the cell-type difference is falsely treated as the omics difference and accordingly eliminated, the cells of different types would be integrated, leading to the over-integration problem. On the other hand, when the omics difference is falsely treated as the cell-type difference and insufficiently reduced, the cells of different omics would not be well mixed, leading to the under-integration problem.

Here, we reveal that instead of being an interference, the cell heterogeneity could be exploited to facilitate data integration based on the following observation. Specifically, the chromatin accessibility of scATAC-seq cells exhibits variable correlations with gene expression of scRNA-seq[28,29]. scATAC-seq cells with higher positive correlation exhibit smaller omics differences, which are easier to integrate and could bridge the modality gap between the two omics. According to the observation, we designed scBridge, a heterogeneous transfer learning method for multi-omics data integration. Briefly, scBridge first warms up a deep neural classifier with the annotated scRNA-seq data, and then identifies the scATAC-seq cells with smaller omics differences through reliability modeling. After that, the reliable scATAC-seq cells are integrated with scRNA-seq cells through cross-omics prototype alignment. Lastly, scBridge selects and merges the most reliable scATAC-seq cells into the annotated scRNA-seq data to narrow the omics gap. By repeating the above processes, the omics difference would be gradually reduced, and more cells would be integrated, leading to the final integration result.

We evaluate the data integration performance of scBridge on seven multi-omics datasets in terms of joint embedding quality and label transfer accuracy. Extensive experimental results illustrate the superiority of scBridge in data integration compared with scJoint[15] (semi-supervised), Seuart[22] (MNN-based), Portal[25] (adversarial), Harmony[14] (transcriptomic-oriented), GLUE[24] and Conos[23] (graph-based). Furthermore, the empirical evaluations show that scBridge is robust against the number and quality of scRNA-seq annotations, the inconsistency between scRNA-seq and scATAC-seq cell types, and technical noises in sequencing data.

## Results

### The scBridge algorithm

scBridge is a semi-supervised method that integrates the annotated scRNA-seq data and the unlabeled scATAC-seq data in a heterogeneous transfer learning manner. As illustrated in Fig. 1, scBridge passes the data into a deep neural encoder and a classifier to achieve data integration and label transfer with the help of a reliability modeling module (Overview **a**). To be specific, scBridge first warms up the networks using the annotated data (Step **b**). After that, the networks are transferred to scATAC-seq data. However, such a vanilla transfer paradigm would misclassify scATAC-seq cells due to the modality gap between RNA and ATAC omics. Hence, to integrate the cells of

different types correctly, heterogeneous transfer learning is proposed by utilizing cell heterogeneity. Specifically, a portion of scATAC-seq cells exhibits smaller omics differences with scRNA-seq cells as their chromatin accessibility has higher positive correlations with gene expression. Consequently, the classification results on those scATAC-seq cells are more reliable. To estimate the reliability of each scATAC-seq cell, scBridge models the discriminability and confidence of scATAC-seq cells with a Gaussian Mixture (Step **c**). To be specific, scBridge computes the discriminability of each scATAC-seq cell based on its distance to the RNA prototypes (computed by averaging scRNA-seq cells of different types), as well as the confidence based on its classification loss value. Cells with higher discriminability and confidence are considered more reliable. With the estimated cell reliability, scBridge computes the ATAC prototypes as the weighted average of scATAC-seq cells with the same predicted cell type and aligns them with the RNA prototypes to achieve integration (Step **d**). Lastly, scBridge selects the most reliable scATAC-seq cells and merges them into the annotated data, with labels given by the current classification results (Step **e**). The selected scATAC-seq cells could act as a bridge to reduce the modality gap between RNA and ATAC omics. By repeating steps **b** to **e**, scBridge takes a from-easy-to-hard learning fashion to further identify and integrate the remaining scATAC-seq cells that have more distinct features with scRNA-seq cells.

### scBridge achieves promising integration results on the golden benchmarks

To evaluate the integration performance of scBridge, we first applied it to three golden benchmarks including the SNARE-seq dataset of mouse brain cortex[18], the SHARE-seq dataset of human bone marrow[30], and the 10x Multiome dataset of mouse kidney[31]. As these three sequencing techniques could link the cell's transcriptome with its accessible chromatin, the pairing information provides a golden criterion to validate the integration performance. Notably, the pairing information was not used during integration, but only for validation. Moreover, these three datasets cover three different tissues and two species, which also evaluates the generalization ability of the methods.

To intuitively show how scBridge iteratively integrates scRNA-seq and scATAC-seq data through heterogeneous transfer learning, we visualized the integration process on the SNARE-seq dataset in Fig. 2c. To be specific, the right figure shows the Pearson correlation score (computed on all genes) between scRNA-seq cells and the selected reliable scATAC-seq cells, where larger scores denote smaller omics differences between scRNA-seq and scATAC-seq cells. As shown, scBridge first integrates scATAC-seq cells that are most similar to scRNA-seq cells and gradually integrates more distinct ones in the subsequent iterations ($t$-test $p$-value $< 1e − 3$ in the first five iterations, with the Pearson correlation score decreasing significantly). Such a trend also holds in different types of cells as illustrated in Supplementary Fig. 1a. Here, we took Ex-L2/3-Rasgrf2 cells of scATAC-seq as an example to demonstrate various cell correlation levels across omics. Supplementary Fig. 1b shows a decrease in the *Rasgrf2* gene activity as the model iterates. Meanwhile, Supplementary Fig. 1c demonstrates that the discrepancy between *Rasgrf2* gene activity and gene expression increases as integration proceeds, consistent with the results in Supplementary Fig. 1a. The left figure in Fig. 2c demonstrates the reliable cell selection and overall label transfer accuracy across iterations. In brief, after the first iteration, scBridge achieves 60.11% label transfer accuracy for all scATAC-seq cells. Based on the Gaussian Mixture Model, 1898 scATAC-seq cells are selected as the annotated data with an accuracy of 90.89%. By using those reliable scATAC-seq cells to bridge RNA and ATAC omics, scBridge achieves better integration results (63.62% label transfer accuracy) in the second iteration. As the training proceeds, more scATAC-seq cells are selected as reliable by scBridge, and the label transfer accuracy steadily grows to 71.95%.

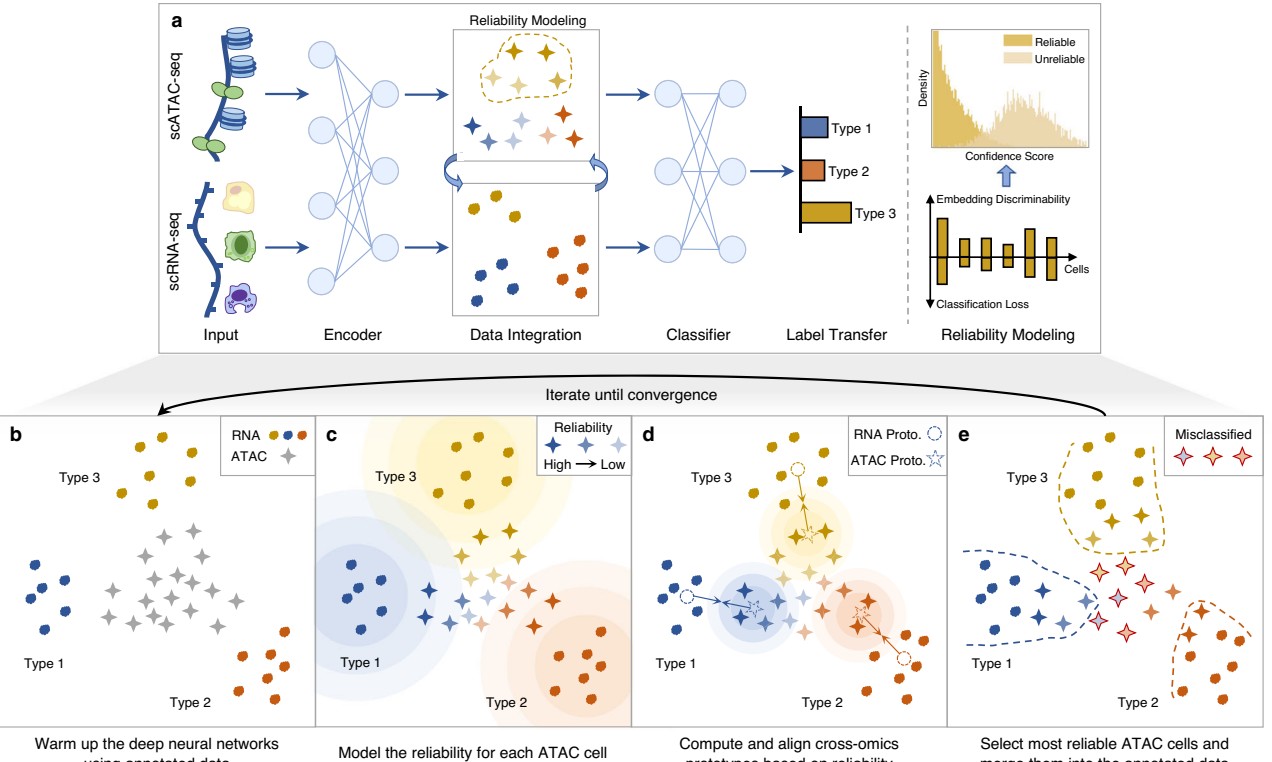

**Fig. 1 | Overview of scBridge. a** The input of scBridge is composed of the annotated scRNA-seq data and unlabeled scATAC-seq data. scBridge passes them into a deep neural encoder and a classifier to achieve data integration and label transfer iteratively. The main steps of scBridge are elaborated as follows. **b** scBridge warms up the deep neural encoder and classifier with the annotated scRNA-seq data and accordingly obtains the initial embedding and cell-type prediction for scRNA-seq and scATAC-seq cells. **c** scBridge models the reliability of each scATAC-seq cell based on its embedding discriminability and classification loss. **d** Based on the

estimated reliability, scBridge computes the ATAC prototypes as the weighted average of scATAC-seq cells and aligns them with the RNA prototypes for integration. **e** At the end of each iteration, scBridge selects the most reliable scATAC-seq cells and merges them into the annotated data, with labels given by the current predictions. scBridge repeats steps **b** to **e** until convergence so that more and more ATAC cells would be integrated, leading to the final data integration and label transfer results.

Figure 2a and Supplementary Fig. 2a illustrate the final data integration results achieved by scBridge and six baseline methods. As shown, though all seven methods successfully mix scRNA-seq and scATAC-seq cells, scBridge and scJoint achieve more discriminative cell clusters compared with other baselines. In some clusters, however, scJoint falsely integrates cells with different types, leading to inferior label transfer performance. To further validate the superiority of scBridge, Fig. 2d and Supplementary Fig. 2b visualize the confusion matrix of the transferred labels. The results show that scBridge discriminates the cells of different types more accurately compared with all baselines. For example, scJoint fails to separate Claustrum, Mic, and OPC cells, whereas scBridge achieves almost perfect label transfer on them. By using the silhouette score and label transfer accuracy to quantitatively evaluate the integration results, Fig. 2b shows that scBridge achieves the highest harmonized silhouette score, indicating its superiority in the removal of omics difference and the preservation of cell-type difference. We also noticed that scBridge achieves a more precise integration for the rare cell types, i.e., a significant improvement on the weighted F1-score in label transfer (42.26% by scBridge compared with 22.12% by scJoint).

We further visualized the joint embeddings obtained by scBridge and scJoint on the SHARE-seq and 10x Multiome datasets in Fig. 2e–f. On the SHARE-seq dataset, scBridge achieves better cell grouping than scJoint, especially for the rare types like Baso. On the 10x Multiome dataset, scBridge successfully mixes scRNA-seq and scATAC-seq cells, while scJoint fails to eliminate the gap between the two modalities. The UMAP visualizations, label transfer matrix, and quantitative metrics of scBridge and all other baselines are in Supplementary Figs. 3–4

demonstrate the superior performance of scBridge in data integration and label transfer.

Finally, as the heterogeneous transfer learning paradigm of scBridge requires the annotated scRNA-seq data, a natural question is how many annotated scRNA-seq cells are needed for accurate integration. To answer this question, we evaluated the robustness of scBridge against the number of annotations on the three golden benchmarks, compared with four baselines that support label transfer. Specifically, we carried out experiments by using 100%, 75%, 50%, and 25% of scRNA-seq data. Figure 2g shows that scBridge achieves the best label transfer accuracy and F1-score under all downsample rates on three benchmarks. Notably, on the 10x Multiome dataset, scBridge remains a high average F1-score of 77.08% with only 25% annotated scRNA-seq cells compared with 77.35% on full data. In contrast, scJoint encounters a significant performance drop in average F1-score, i.e., from 73.22% on full data to 59.36% on 25% downsampled data ($t$-test $p$-value $= 1.94e-5$, degrees of freedom $= 8$, 95% confidence interval $= [0.103, 0.174]$). In addition, scBridge with only 50% scRNA-seq annotations outperforms all baselines with full data on the SNARE-seq dataset. Such a data-efficient property of scBridge could be attributed to its heterogeneous transfer learning paradigm. Namely, as long as the annotated scRNA-seq data is enough for identifying a portion of reliable scATAC-seq data, scBridge could progressively integrate the rest cells.

## scBridge scales to atlas data
With the development of sequencing techniques, the number of cells profiled with various protocols grows continually, arousing the

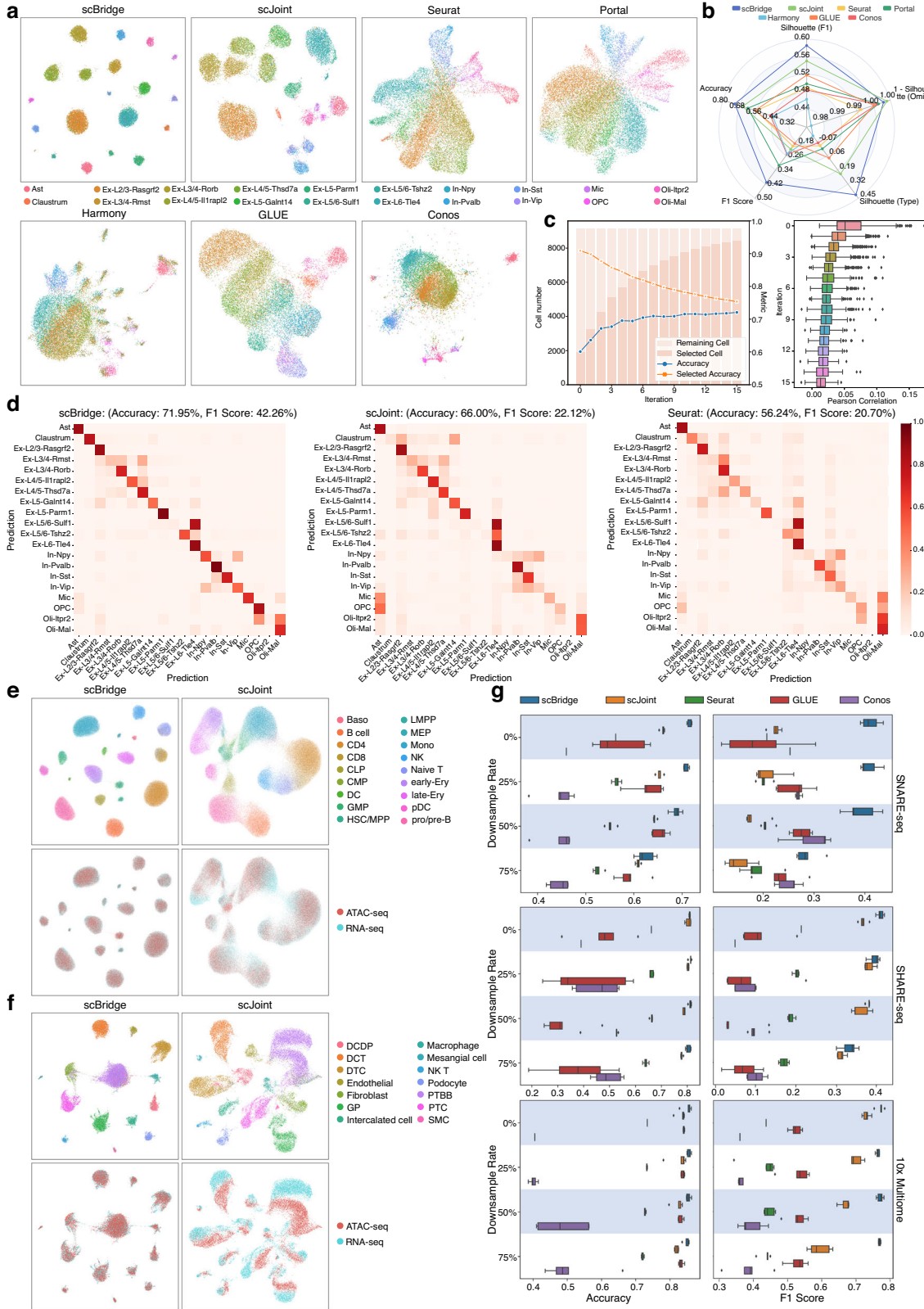

demand for efficiently handling large-scale data. To access how scBridge scales to large data, we evaluated it on the mouse atlas dataset. Specifically, we used the cells sequenced with FACS and droplet protocols provided by Tabula-Muris[32] as scRNA-seq data, and the cells sequenced by Cusanovich et al.[33] as scATAC-seq data. After data preprocessing, 102,103 cells from 18 common types are selected for evaluation.

To investigate the computation efficiency of scBridge, we applied it to five subsets of mouse atlas with 5,000–80,000 cells. Figure 3c shows the (logged) running time and memory consumption of all tested methods with respect to different cell numbers. As shown, scBridge takes linearly increasing running time (the third-best) and constant memory consumption (the second-best), which is favorable in scaling to large data.

**Fig. 2 | Integration results on three golden benchmarks. a** UMAP plot of the joint embeddings obtained by the seven methods on the SNARE-seq dataset, where cells are colored by types. **b** Quantitative evaluation on the SNARE-seq dataset in terms of the joint embedding quality and label transfer accuracy. Source data are provided as a Source Data file. **c** (Left) The number of reliable scATAC-seq cells selected by scBridge with the corresponding accuracy, and the overall label transfer accuracy across the training process on the SNARE-seq dataset. (Right) The Pearson correlation score between scRNA-seq and the selected scATAC-seq cells in different iterations. Source data are provided as a Source Data file. **d** The label transfer matrix of the agreement between the predicted cell type and the ground-truth annotation. A clearer diagonal structure denotes better label transfer performance. **e** UMAP

embeddings of scBridge and scJoint on the SHARE-seq dataset. **f** UMAP embeddings of scBridge and scJoint on the 10x Multiome dataset. The first and second rows show cells colored by types (DCDP: Distal collecting duct principal cell, DCT: Distal convoluted tubule cell, DTC: Distal tubule cell, PTC: Proximal tubule cell, SMC: Smooth muscle cell, GP: Glomerular podocyte, PTBB: Proximal tubule brush border cell) and omics, respectively. **g** The label transfer accuracy and F1-score of the tested methods on three benchmarks, where 100%, 75%, 50%, and 25% annotated scRNA-seq data are used. Five random experiments are conducted with different downsample rates. Each boxplot ranges from the upper and lower quartiles with the median as the horizontal line and whiskers extend to 1.5 times the interquartile range. Source data are provided as a Source Data file.

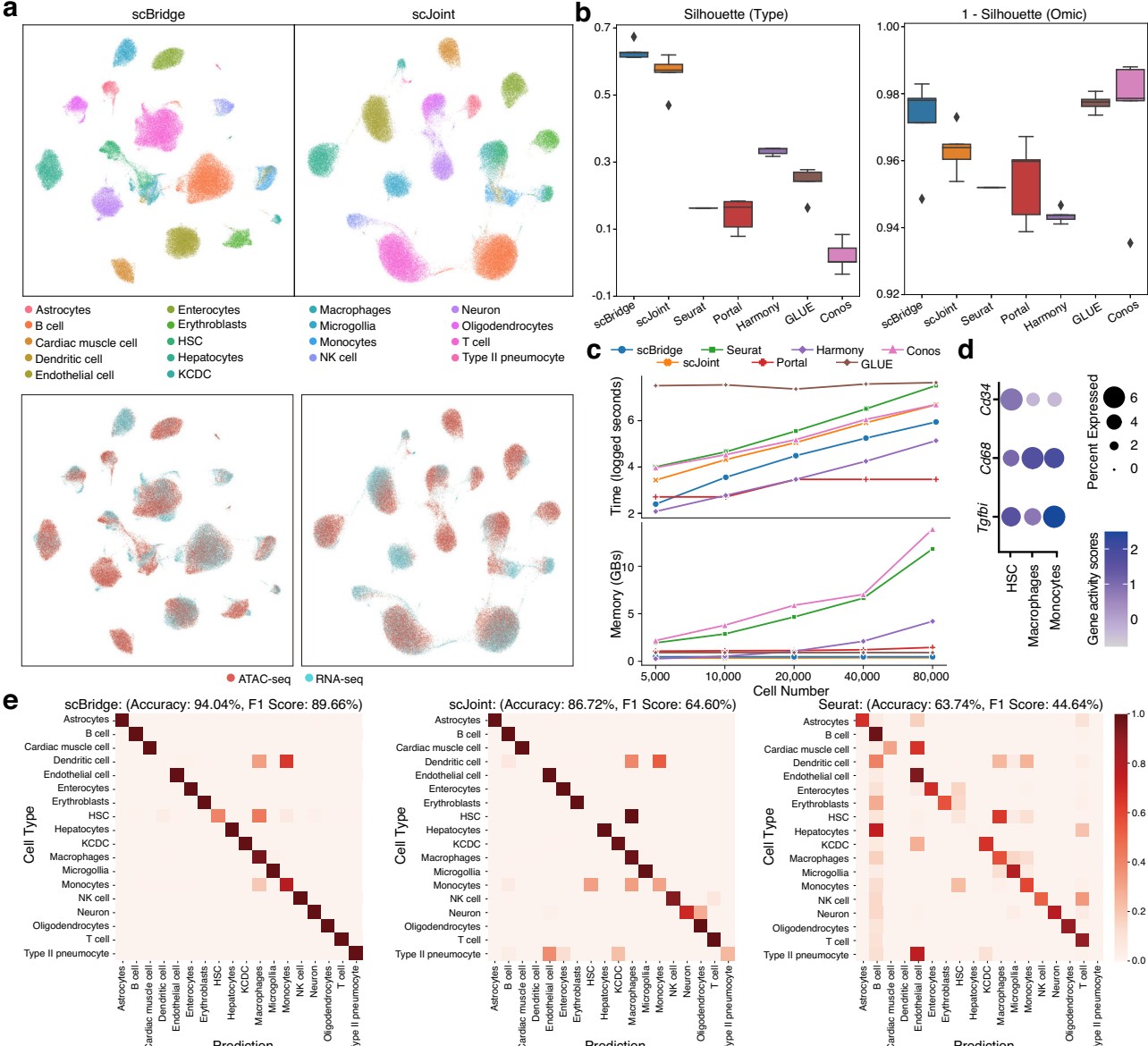

**Fig. 3 | Integration results on mouse atlas data. a** UMAP visualization of the joint embeddings learned by scBridge and scJoint. The first and second rows show cells colored by types (HSC: Hematopoietic stem cell, KCDC: Kidney collecting duct cell) and omics, respectively. **b** The cell type and 1 − omics silhouette coefficients of scBridge and six baselines on five random experiments. Each boxplot ranges from the upper and lower quartiles with the median as the horizontal line and whiskers extend to 1.5 times the interquartile range. Source data are provided as a Source Data file. **c** The running time and memory consumption of different methods on

mouse atlas subsets of 5,000--80,000 cells. Source data are provided as a Source Data file. **d** The dot plot of relative expression of marker genes *Tgfbi, Cd68, Cd34* in cells predicted as HSC, Macrophages, and Monocytes by scBridge. The size of the circle represents the proportion of expressing cells, and the color indicates the average expression level. **e** The agreement between the predicted label and the manual annotation. Matrices with a clearer diagonal structure indicate better performance.

Despite the scalability and efficiency of scBridge, we also evaluated its effectiveness on the full mouse atlas data. As visualized in Fig. 3a and Supplementary Fig. 5a, scBridge achieves better grouping of cells by types and mixing of cells by omics. The superiority of scBridge is also verified according to the cell type and omics silhouette score in Fig. 3b. To evaluate the label transfer performance, we illustrated the label transfer matrix in Fig. 3e and Supplementary Fig. 5b. As shown, scBridge (89.66% F1-score) achieves a clearer diagonal label transfer matrix compared with scJoint (64.60% F1-score) and Seurat (44.64% F1-score), which indicates a more precise cell integration. Additionally, we observed that Hematopoietic stem cells (HSC) were annotated as HSC, Macrophage, and Monocyte by scBridge, but only Macrophage by scJoint. To explore such a difference, we computed the activity score of the marker genes *Cd34*, *Cd68*, *Tgfbi* for the three types in Fig. 3d. The results verify that scBridge makes more reasonable and accurate cell-type predictions than scJoint. Notably, despite the immense differences between FACS and droplet data (the FACS method captures fewer cells but detects more molecules per cell than the microfluidic-droplet method), scBridge does not require any pre-integration of these two batches of data, which demonstrates its capacity to handle data with batch effects. Note that not all results agree with those reported in the scJoint paper[15] due to the slight differences in data preprocessing.

To further investigate the influence of the annotation number, we carried out experiments by using 25%, 50%, and 75% of scRNA-seq data. As shown in Supplementary Fig. 5c, even with only 25% annotated scRNA-seq cells, scBridge still outperforms scJoint with 100% data ($t$-test $p$-value = 0.014, degrees of freedom = 8, 95% confidence interval = [0.010, 0.066]). In addition, the weighted F1-score gaps between scBridge and the baselines are significant, indicating the superiority of scBridge in integrating cells of rare types. For example, under the downsample rate of 75%, scBridge outperforms scJoint by 25.03% ($t$-test $p$-value = 2.84$e$ − 5, degrees of freedom = 8, 95% confidence interval = [0.182, 0.318]) and Seurat by 47.36% ($t$-test $p$-value = 3.23$e$ − 9, degrees of freedom = 8, 95% confidence interval = [0.434, 513]) in term of F1-score on average. In summary, the computational- and data-efficient properties, as well as its superior performance, make scBridge a promising tool in handling large-scale multi-omics data.

## scBridge handles data with inconsistent cell type across omics

In the above experiments, the cell type is consistent across scRNA-seq and scATAC-seq data. In practice, however, such consistency does not always hold. Hence, it is highly expected to explore how data integration methods behave when cell types unmatch across omics. For this purpose, we evaluated scBridge on the human myocardial infarction data[34], which consists of 67,360 scRNA-seq cells from 11 types, and 46,086 scATAC-seq cells from 8 types after preprocessing. More specifically, Mast, Adipocyte, and Cycling cells are only observed in scRNA-seq data. Experimental results in Supplementary Fig. 7 show that scBridge not only learns the joint embedding with better cell-type grouping and omics mixing but also achieves the best performance in all five data integration and label transfer metrics.

Furthermore, we conducted a more challenging evaluation by manually removing the Myeloid cells from scRNA-seq data. In other words, there are only 7 cell types in common for scRNA-seq and scATAC-seq data, and both of them have unique cell types. The UMAP visualizations in Fig. 4a and Supplementary Fig. 6a illustrate that scJoint fails to integrate the cells from different omics, and other methods achieve less distinct partition of cells with different types compared with scBridge. According to the label transfer matrix in Fig. 4d, scBridge, and Seurat transfer fewer scATAC-seq cells of common types to the three unique types in scRNA-seq data than GLUE and Conos, and scBridge achieves more precise label transfer results among the seven common types. Next, we focused on the integration results for Myeloid cells in scATAC-seq data, which is novel with

respect to the annotations in scRNA-seq data. Equipped with the structure preservation loss, scBridge assigns a relatively low confidence score for scATAC-seq Myeloid cells as shown in Fig. 4b. To identify cells of novel types, instead of manually setting a confidence threshold, we proposed a data-driven strategy by fitting the confidence score of all cells with a two-component GMM. As shown in Fig. 4b, the confidence threshold is estimated by the intersection of two probability density functions (PDF). In other words, cells belonging to the less confident GMM component are considered novel. According to the novel type identification performance shown in Fig. 4c and Supplementary Fig. 6d, scBridge gives a more distinct pattern between cells of common and novel types, leading to the highest F1-score for novel type discovery. The superiority of scBridge is also verified by the label transfer matrix in Fig. 4d, namely, it assigns fewer cells of common types as novel.

## scBridge is robust to the dropout technical noise in sequencing data

In single-cell sequencing studies, it is inevitable to introduce technical noises in sequencing data due to biological and technical limitations such as amplication bias, low starting mRNA amount, and sequencing depth. For example, a typical technical noise is the dropout event, where some entries in the gene expression or activity matrices are false-zeros[35]. Accordingly, the data would be contaminated with considerable non-biological variances, hurting the data integration performance.

To investigate the robustness of scBridge against the dropout technical noise, we applied it to the human hematopoiesis data which contains 34,609 scRNA-seq and 33,819 scATAC-seq cells from 23 common types. To simulate the dropout events, we downsampled the scRNA-seq count matrix, scATAC-seq activity matrix, and scATAC-seq peak-by-cell matrix by 25%, 50%, and 75%, respectively, with the downsampleMatrix function provided in the scuttle R package[36]. As shown in Fig. 5c, scBridge achieves superior robustness towards the scRNA-seq data quality. Namely, its integration and label transfer performances are almost impervious under up to 75% dropout rate on scRNA-seq data. By comparison, though GLUE achieves higher label transfer accuracy than scBridge on the original data, its performance becomes worse and unstable on data contaminated with dropout noises. Similarly, scJoint achieves a comparable silhouette score with scBridge, but encounters prominent performance reduction as the dropout rate increases. Figure 5a, b, and Supplementary Fig. 8 demonstrate the superiority of scBridge over six baselines in data integration and label transfer. Likewise, scBridge also achieves better performance on the corrupted scATAC-seq data, especially under high dropout rates as shown in Fig. 5c. Such robustness of scBridge could be attributed to its iterative and heterogeneous integration paradigm. Namely, even if the sequencing data is of low capture rate, scBridge could still identify a portion of reliable scATAC-seq data, which further helps the model to integrate the rest cells. Note that some results do not exactly match those reported in the scJoint paper[15] due to the differences in data preprocessing and the added dropout corruption.

## scBridge is robust to noisy labels in scRNA-seq data annotation

Cell-type annotation is challenged by incomplete messenger RNA detection, a lack of curated marker gene lists, improper handling of batch effects, and difficulties in leveraging the latent gene-gene interaction information[37]. It is inevitable to introduce some noisy labels during the manual or automatic annotation. As scBridge requires annotated scRNA-seq data, it is highly expected that scBridge is robust against noisy labels.

We conducted evaluations on the multi-modal PBMC data[38] consisting of 4644 and 4157 cells of seven types from scRNA-seq and scATAC-seq data, respectively. As shown in Fig. 6b, c, scBridge, scJoint, Seurat, and GLUE achieve promising results when the annotations in

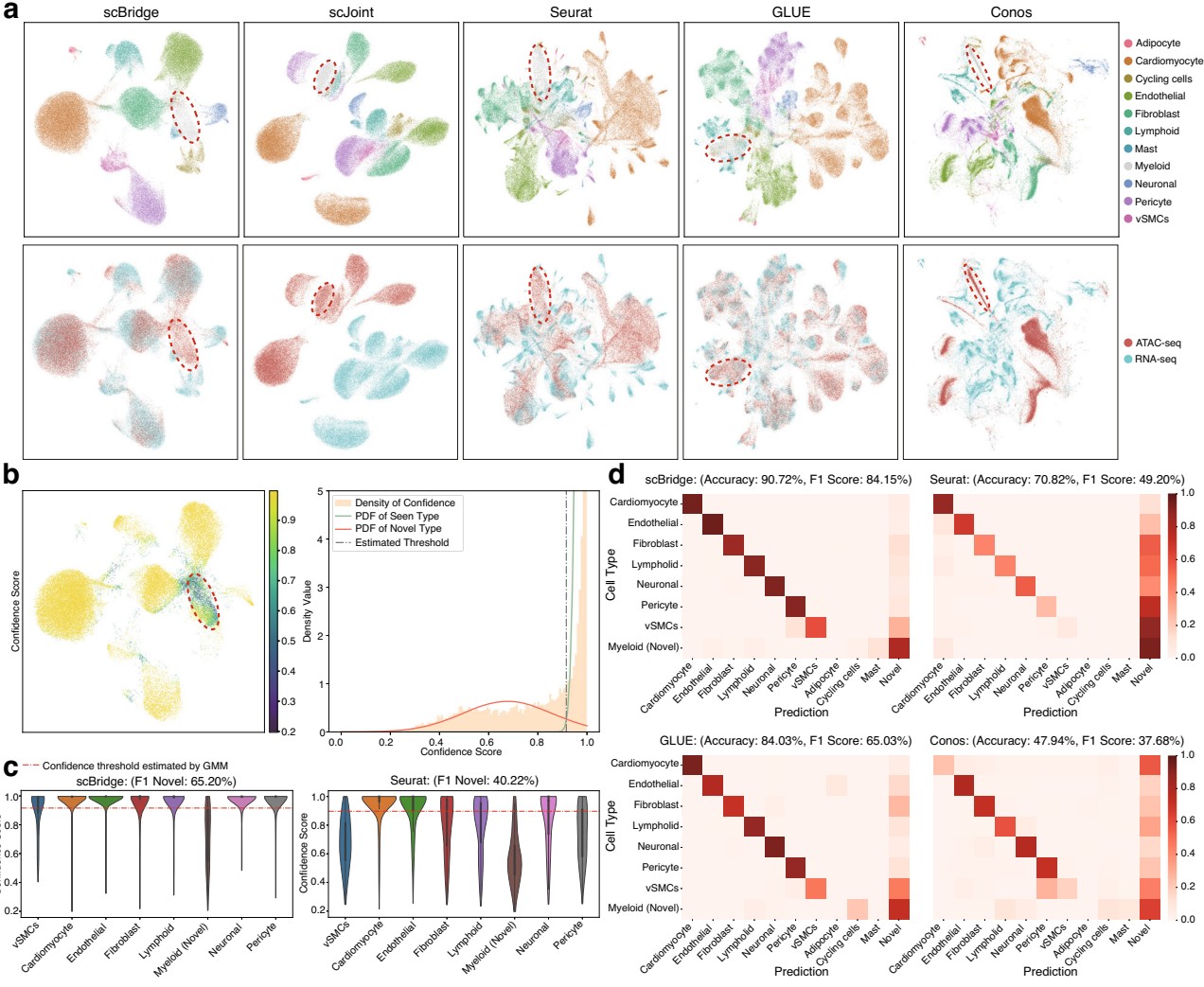

**Fig. 4 | Integration results on the human myocardial infarction data, where both RNA and ATAC omics have their unique cell types (adipocyte, cycling cells, and mast cells only exist in scRNA-seq data, and Myeloid cells only exist in scATAC-seq data). a** UMAP visualization of the joint embedding obtained by scBridge, scJoint, Seurat, GLUE, and Conos. The first and second rows show cells colored by types and omics, respectively. The novel Myeloid cells are gray-colored and red-circled. **b** (Left) scBridge's UMAP embedding of scATAC-seq cells, colored by the confidence score. (Right) scBridge's novel type threshold was estimated by applying a two-component GMM on the confidence score. **c** The confidence score predicted by scBridge and Seurat on different types of 46,086 scATAC-seq cells. The red dashed line corresponds to the novel type confidence threshold estimated by GMM. Each miniature boxplot ranges from the upper and lower quartiles with the median as the horizontal line and whiskers extend to 1.5 times the interquartile range. **d** The label transfer results of scBridge, Seurat, GLUE, and Conos. Cells are considered novel if their confidence scores are below the threshold estimated by GMM.

scRNA-seq data are accurate. Specifically, scBridge achieves the best label transfer accuracy and F1-score, outperforming the second-best method scJoint by about 5% on average with *t*-test *p*-value ≤ 1e-3 in the two metrics. To simulate label corruptions, we randomly shuffled 5%, 10%, and 20% percent of annotations in the scRNA-seq data. According to the results in Fig. 6c, scJoint achieves inferior performance than Seurat and GLUE even under a small label corruption rate of 5%. In contrast, scBridge outperforms them under all corruption rates (*t*-test *p*-value = 2e-6, 8e-4, and 0.1 under rates 5%, 10%, and 20% respectively in F1-score compared with Seurat), demonstrating its ability to handle data with label corruptions.

Finally, we validated scBridge's cell-type predictions of Naive CD4+ T cells using Protein CD45RA and gene *CCR7*, as well as Effector CD4+ T cells using Protein CD45RO and gene *PLEKHG3* in Fig. 6d. Furthermore, we reconstructed the cell differentiation trajectory using monocle[39] in Supplementary Fig. 9c. As shown, the RNA Pseudotime shows a trend from Naive CD4+ T cells to Effector CD4+ T cells, reflecting the CD4+ T cell differentiation. The UMAP plots in Fig. 6d

show that scBridge reflects the continuous trajectory better than scJoint.

## Discussion

By utilizing cell heterogeneity, scBridge achieves accurate scRNA-seq and scATAC-seq data integration, as well as label transfer with heterogeneous transfer learning. To summarize, scBridge accepts annotated scRNA-seq data and unlabeled scATAC-seq data to perform integration in an iterative manner. In each iteration, scBridge models the reliability of heterogeneous scATAC-seq cells and conducts cross-omics prototype alignment based on the estimated cell reliability. After that, scBridge selects the most reliable scATAC-seq cells as the annotated data and repeats the entire process. As the training proceeds, the modality gap between RNA and ATAC omics is gradually reduced, which enables scBridge to identify and integrate more scATAC-seq cells, leading to the final integration result. On seven multi-omics data integration benchmarks, scBridge outperforms six representative baselines in both joint embedding quality and label

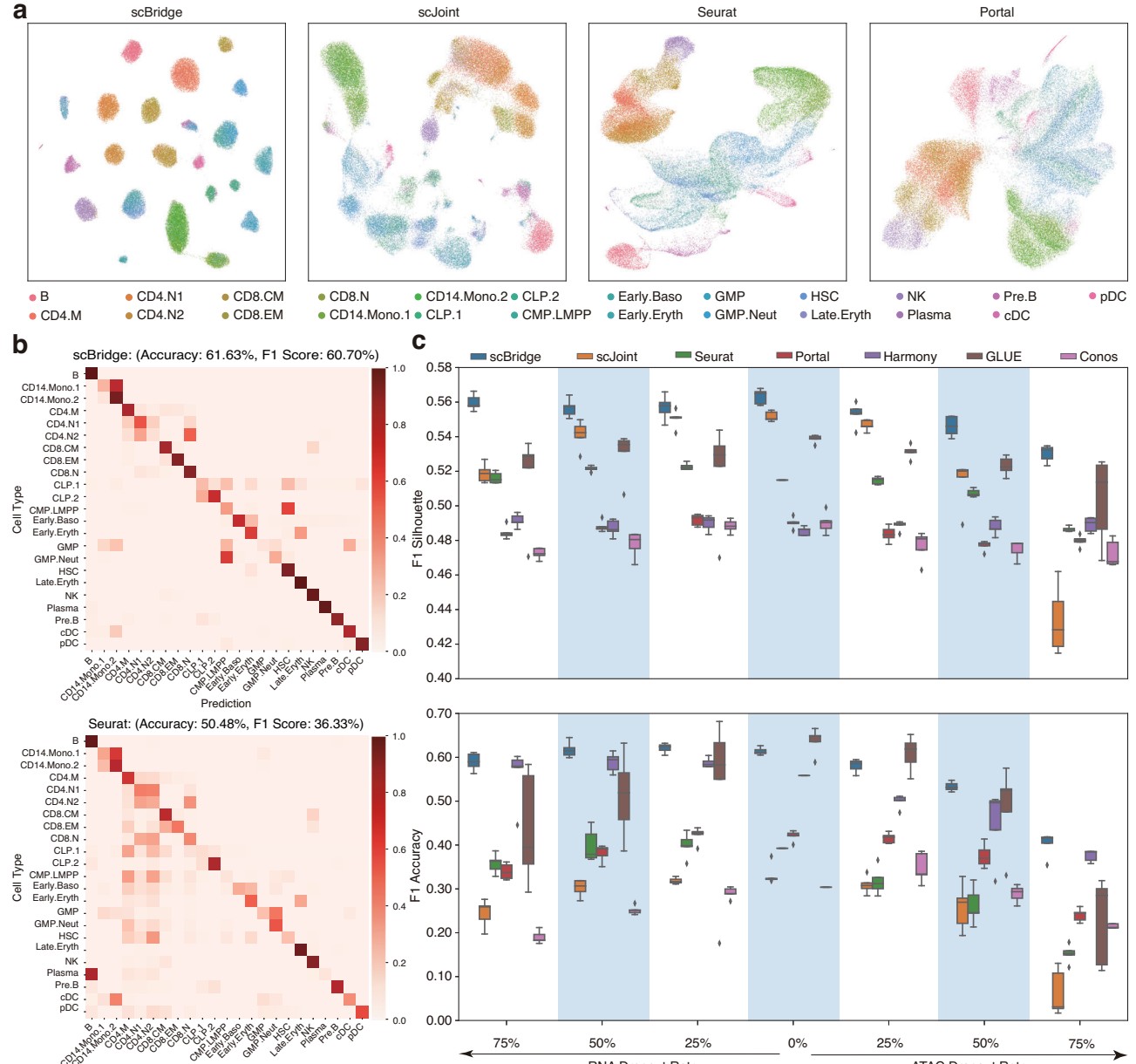

**Fig. 5 | Integration results on human hematopoiesis data. a** UMAP visualization of the joint embeddings learned by scBridge, scJoint, Seurat, and Portal under 75% dropout on scRNA-seq data, where cells are colored by types. **b** The agreement between labels transferred by scBridge, Seurat, and the manual annotations under 75% dropout on scRNA-seq data. A clearer diagonal structure indicates better agreement. **c** The F1 harmonized silhouette score and the weighted F1 label transfer accuracy of scBridge and six baselines with different dropout corruption rates on scRNA-seq and scATAC-seq data. Five random experiments are conducted under each dropout rate. Each boxplot ranges from the upper and lower quartiles with the median as the horizontal line and whiskers extend to 1.5 times the interquartile range. Source data are provided as a Source Data file.

transfer accuracy. In addition to its superior performance, scBridge also shows strong robustness against (i) the number of annotated scRNA-seq cells, (ii) the inconsistency between scRNA-seq and scATAC-seq cell types, (iii) the dropout technical noise in sequencing data, and (iv) the quality of scRNA-seq annotations.

Though scBridge is a deep learning-based method, we simplified its structure and hyper-parameters to avoid laborious parameter tuning on different datasets. We fixed the same set of hyper-parameters on all seven datasets and scBridge achieves the best performance without any parameter tuning. In other words, users only need to decide whether to turn on the structure preservation loss to enable novel type discovery or strictly integrate all scATAC-seq cells with annotated scRNA-seq data, based on practical needs. Moreover, as

scBridge only requires mini-batch optimization, it naturally scales to large-scale data, with a linear time and constant memory consumption with respect to the cell number.

In this paper, we have focused on integrating scRNA-seq and scATAC-seq data. But theoretically, scBridge could extend to other modalities as long as the input data matrix is aligned in columns (i.e., genes, proteins, etc.). We found that when directly applied to the protein data from the human peripheral blood mononuclear dataset, scBridge still achieves better integration and label transfer performance compared with scJoint (Supplementary Note 1). Notably, scBridge does not strictly require all scRNA-seq cells to be annotated and scATAC-seq cells to be unlabeled. Cell annotations from any omics could be easily incorporated into the

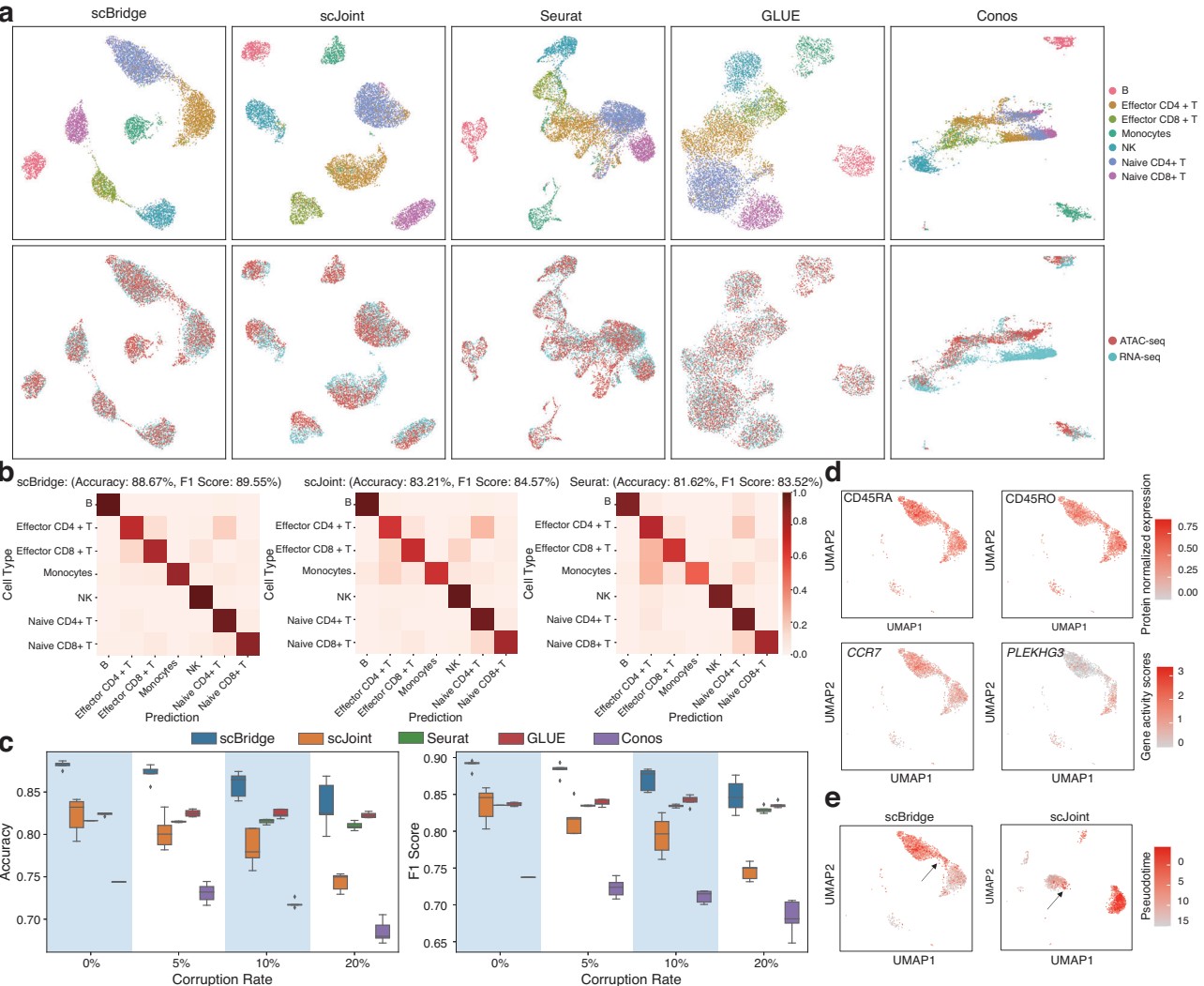

**Fig. 6 | Integration results on human PBMC data. a** UMAP visualization of the joint embeddings obtained by scBridge, scJoint, Seurat, GLUE, and Conos. The first and second rows show cells colored by types and omics, respectively. **b** The label transferred by scBridge, scJoint, and Seurat. A clearer diagonal structure indicates better agreement between the transferred labels and manual annotations. **c** The label transfer accuracy and F1-score of the tested methods with 0%, 5%, 10%, and 20% corruption rates on the scRNA-seq cell annotations. Five random experiments are conducted with different corruption rates. Each boxplot ranges from the upper

and lower quartiles with the median as the horizontal line and whiskers extend to 1.5 times the interquartile range. Source data are provided as a Source Data file. **d** Protein CD45RA, CD45RO and gene *CCR7*, *PLEKHG3* projected on the scBridge UMAP plot of Effector CD4+ T and Naive CD4+ T cells. **e** Pseudotime projected on the scBridge and scJoint UMAP plots of Effector CD4+ T and Naive CD4+ T cells. The arrows indicate continuously changed cells from Naive CD4+ T cells to Effector CD4+ T cells.

heterogeneous integration framework of scBridge, which is flexible in practice.

In conclusion, scBridge is a multi-omics data integration method based on a novel paradigm, *i.e.*, heterogeneous transfer learning. Considering the large amounts of well-annotated scRNA-seq data and a wide range of scRNA-seq annotation tools[40], scBridge has a promising application prospect. With its superior performance compared with existing baselines, robustness against different occasions, and scalability to large datasets, scBridge would be a reliable tool in single-cell multi-omics analysis.

## Methods
### scBridge
For the given scRNA-seq data $X^s \in \mathbb{R}^{n^s \times m}$ and scATAC-seq data $X^t \in \mathbb{R}^{n^t \times m}$, scBridge employs a shared deep encoder network $f: X \to E$ to learn cell embeddings followed by a shared classification head $g: E \to Y$ to classify both $X^s$ and $X^t$, where $n^s$ and $n^t$ denote the number of scRNA-seq and scATAC-seq cells, $m$ is the number of

their common genes, and $X^s \in \mathbb{R}^{n^s \times m}$ is with the annotation $Y^s \in \mathbb{R}^{n^s}$. scBridge integrates multi-omics data in an iterative manner. First, scBridge warms up $f(\cdot), g(\cdot)$ using annotated scRNA-seq data. The performance of $f(\cdot), g(\cdot)$ is limited when directly transferred to scATAC-seq data due to the omics difference. Thanks to cell heterogeneity, a portion of scATAC-seq cells exhibit smaller omics differences with scRNA-seq cells. Consequently, the model could learn more discriminative embeddings and make more accurate cell-type predictions for those scATAC-seq cells. To identify those reliable scATAC-seq cells, scBridge models the reliability of scATAC-seq cells by fitting the embedding discriminability and classification confidence with the Gaussian Mixture. Based on the estimated cell reliability, scBridge computes the prototypes in ATAC omics and aligns them with the RNA prototypes for integration. Lastly, scBridge selects the most reliable scATAC-seq cells as the annotated data and repeats the entire training process. During the iterations, the modality gap between RNA and ATAC omics is gradually narrowed, enabling scBridge to identify and integrate

more cells, and precisely predict their cell type $Y^t \in \mathbb{R}^{n^t}$. The training procedure of scBridge is elaborated below.

**Warm-up with annotated scRNA-seq data.** To endow the deep neural networks $f(\cdot), g(\cdot)$ with the capacity of feature extraction and cell classification, we first use the annotated scRNA-seq data $\{X^s, Y^s\}$ to warm-up $f(\cdot)$ and $g(\cdot)$ with the following weighted cross-entropy loss:

$$L_{WCE} = \frac{1}{N} \sum_{i=1}^{N} -w_i^s \log \left( \frac{\exp(p_i^s[y_i^s])}{\sum_{k=1}^{K} \exp(p_i^s[k])} \right),$$
$$w_i^s = \frac{K/|Y_{y_i}^s|}{1/|Y_{y_1}^s| + \cdots + 1/|Y_{y_K}^s|}, p_i^s = g(f(x_i^s)), \tag{1}$$

where $N$ and $K$ denote the mini-batch size and cell-type number, $x_i^s$ and $y_i^s$ denote the gene expression and annotation of the $i$-th cell, $p_i^s$ refers to the predicted soft label of sample $i$, and $|Y_k^s|$ indicates the number of cells belonging to the $k$-th class. Here the weight $w_i^s$ is adopted to alleviate the influence of the highly imbalanced distribution of cells across different types.

**Reliability modeling for scATAC-seq data with the Gaussian Mixture.** After the warm-up, the model is able to correctly predict cell types for scRNA-seq data. However, the classification performance on scATAC-seq data is undesirable due to the modality gap between RNA and ATAC omics. Nevertheless, thanks to cell heterogeneity, we observe that a portion of scATAC-seq cells exhibit smaller omics differences with scRNA-seq cells when their chromatin accessibility has higher positive correlations with gene expression, and those cells are easier to integrate. Specifically, the deep embedding network $f(\cdot)$ tends to extract more discriminative features, and the classifier $g(\cdot)$ tends to make more confident predictions for those cells. Motivated by such an observation, to identify those reliable cells that have smaller omics differences, we model the reliability of scATAC-seq cells with Gaussian Mixture based on their discriminability and confidence. Specifically, the discriminability $d_i^t$ of scATAC-seq data embedding $E^t = f(X^t)$ is calculated by the distance to class centers of scRNA-seq data embedding:

$$d_i^t = \max \left[ \cos(c_1^s, e_i^t), \ldots, \cos(c_K^s, e_i^t) \right],$$
$$c_k^s = \frac{1}{|X_k^s|} \sum_{x_i^s \in X_k^s} f(x_i^s), X_k^s = \{x_i^s | y_i^s = k\}, \tag{2}$$

where $\cos(\cdot, \cdot)$ denotes the cosine similarity, $c_k^s$ is the center of scRNA-seq cells from the $k$-th class $X_k^s$ in the embedding space, and $e_i^t$ refers to the embedding of the $i$-th scATAC-seq cell. The confidence of each prediction is evaluated through the following cross-entropy loss:

$$l_i^t = -\log \left( \frac{\exp(p_i^t[\hat{y}_i^t])}{\sum_{k=1}^{K} \exp(p_i^t[k])} \right), p_i^t = g(f(x_i^t)), \hat{y}_i^t = \text{argmax}(p_i^t), \tag{3}$$

where $\hat{y}_i^t$ denotes the current prediction for the $i$-th scATAC-seq cell. Note that $l_i^t$ would be smaller when the prediction is more confident (i.e., closer to one-hot). Given the discriminability $d^t = \{d_1^t, d_2^t, \ldots, d_{n^t}^t\}$, as well as the losses $l^t = \{l_1^t, l_2^t, \ldots, l_{n^t}^t\}$ for all ATAC cells, we fit their distribution using two-component GMM $g^a$ and $g^b$, respectively:

$$g^a(d^t) = \gamma_{c_1^a} \phi(d^t | c_1^a) + \gamma_{c_2^a} \phi(d^t | c_2^a),$$
$$g^b(l^t) = \gamma_{c_1^b} \phi(l^t | c_1^b) + \gamma_{c_2^b} \phi(l^t | c_2^b), \tag{4}$$

where $\gamma_{c_1}, \gamma_{c_2}$ denote the mixture coefficient for components $c_1, c_2$, and $\phi(\cdot | c_1), \phi(\cdot | c_2)$ refer to the probability density. Without loss of generality, we assume that the mean values of two components satisfy

$\mu_{c_1} < \mu_{c_2}$. Based on our observation, the cells belonging to components $c_2^a$ and $c_1^b$ are more likely to be correctly classified. Accordingly, we design the following cross-omics prototype alignment strategy to integrate scRNA-seq and scATAC-seq data.

**Cross-omics prototype alignment with the estimated cell reliability.** To align cells of the same type across different omics, we propose minimizing the pair-wise distance between cross-omics prototypes. The prototype in scRNA-seq data corresponds to the class center as defined in Eq. (2), while the prototype in scATAC-seq data is defined as the weighted mean as follows:

$$c_k^t = \frac{1}{|X_k^t|} \sum_{x_i^t \in X_k^t} p(c_2^a | d_i^t) p(c_1^b | l_i^t) f(x_i^t), X_k^t = \{x_i^t | \hat{y}_i^t = k\}, \tag{5}$$

where $p(c_2^a | d_i^t) \in [0,1]$ corresponds to the probability of cell $i$ belonging to the second component $c_2^a$ in GMM $g^a$, and $p(c_1^b | l_i^t) \in [0,1]$ the probability belonging to the first component $c_1^b$ in GMM $g^b$. Such a GMM-based sample weighting approach is designed to alleviate the influence of false classified cells and prevent the over-integration problem, because the false classified cells are likely to be assigned with relatively low probabilities $p(c_2^a | d_i^t)$ and $p(c_1^b | l_i^t)$.

Our integration objective is to minimize the cosine distance between prototypes $c_k^s$ and $c_k^t$. However, it is daunting to compute global prototypes in the mini-batch optimization paradigm. Hence, as a remedy, we initialize the prototypes with global information and updated them with the exponential moving average via

$$\bar{c}_k^s \leftarrow \eta \bar{c}_k^s + (1-\eta) * c_k^s,$$
$$\bar{c}_k^t \leftarrow \eta \bar{c}_k^t + (1-\eta) * c_k^t, \tag{6}$$

where $\bar{c}_k^{\cdot}$ is globally initialized prototype, $c_k^{\cdot}$ refers to the prototype in each mini-batch, and $\eta$ is the momentum parameter. The data integration is achieved by minimizing the pair-wise distance between cross-omics prototypes:

$$L_{ALN} = \sum_{k=1}^{K} \cos(\bar{c}_k^s, \bar{c}_k^t). \tag{7}$$

**Iterative integration with heterogeneous transfer learning.** Due to the cell heterogeneity and large modality gap between RNA and ATAC omics, it is hard to accurately integrate all cells at a time. As a solution, we take an iterative integration paradigm. To be specific, we select the most reliable scATAC-seq cells $\tilde{X}^t$ as annotated data at the end of each iteration if their GMM probabilities $p(c_2^a | d_i^t)$ and $p(c_1^b | l_i^t)$ are larger than the threshold $\alpha$. After that, we treat the annotated scRNA-seq data and the selected scATAC-seq data as annotated. Together with the remaining unlabeled scATAC-seq data, we retrain the network via

$$X^s \leftarrow X^s \cup \tilde{X}^t,$$
$$X^t \leftarrow X^t \setminus \tilde{X}^t. \tag{8}$$

Such a heterogeneous transfer learning paradigm could gradually eliminate the omics difference, allowing the model to accurately integrate and classify more cells. The above process is repeated until no more reliable scATAC-seq data can be selected.

To sum up, scBridge first warms up the deep neural networks $f(\cdot)$ and $g(\cdot)$ with the annotated scRNA-seq data by Eq. (1). After that, scBridge models the cell reliability with Eq. (4), and performs cross-omics prototype alignment with Eq. (7), namely,

$$L_{Bridge} = L_{WCE} + L_{ALN}. \tag{9}$$

Finally, scBridge selects the most reliable scATAC-seq cells as the annotated data and repeats the training process until convergence.

The overall algorithm of scBridge is summarized in Supplementary Note 4.

**(Optional) Structure preservation for novel-type discovery.** With our iterative integration paradigm, eventually, almost all scATAC-seq cells would be annotated. In other words, they would be integrated with the most similar cells in scRNA-seq data. However, the scRNA-seq annotations may not cover all cell types in scATAC-seq data. To address this challenge, scBridge is equipped with the following structure preservation loss to develop the capacity of novel cell-type discovery, namely,

$$L_{STC} = \frac{1}{N^2} \sum_{i,j} \max \left[ \left( A_{ij}^t - \hat{A}_{ij}^t \right)^2 - \epsilon, 0 \right],$$

$$A_{ij}^t = \begin{cases} 1, & \text{if } x_i^t, x_j^t \text{ are } K \text{ mutual neighbor} \\ 0, & \text{else} \end{cases}, \hat{A}_{ij}^t = \sigma(\hat{x}_i^t \hat{x}_j^{t\top}), \quad (10)$$

where $\epsilon$ is the relaxation parameter which is set to 0.01 by default, $\sigma$ denotes the ReLU activation, $A^t$ is the adjacency matrices computed from the raw scATAC-seq data, and $\hat{A}^t$ is the approximation of $A^t$. Instead of directly computing $\hat{A}^t$ in the embedding space, we adopt a two-layer network $\hat{f}(\cdot)$ to project the embedding $E^t$ into another hidden space via $\hat{x}^t = \hat{f}(E^t)$ wherein $\hat{A}^t$ is computed. Accordingly, the objective function of scBridge becomes:

$$L'_{Bridge} = L_{WCE} + L_{ALN} + L_{STC}. \quad (11)$$

After training, scBridge computes the confidence score $r_i^t$ for each scATAC-seq cell based on the estimated cell reliability, namely,

$$r_i^t = p(c_2^a | d_i^t) p(c_1^b | l_i^t), \quad (12)$$

where $p(c_2^a | d_i^t) \in [0,1]$ corresponds to the probability of cell $i$ belonging to the second component $c_2^a$ in GMM $g^a$, and $p(c_1^b | l_i^t) \in [0,1]$ the probability belonging to the first component $c_1^b$ in GMM $g^b$, following the definitions in Eq. (4) and (5). With the structure preservation loss, scBridge would assign a lower confidence score for the scATAC-seq cells with unseen types, thus enabling novel type discovery. More details and guidance on novel type discovery are provided in Supplementary Note 2. To balance the integration performance and the sensitivity of novel type discovery, we recommend stopping the iteration when more than 75% scATAC-seq cells are selected as the annotated data under such a setting.

**Implementation details.** scBridge is implemented in Python using the PyTorch[41] framework, v.1.12.1. The deep embedding network $f(\cdot)$ is a fully connected network (FCN) with the dimension of m-256-64, where $m$ is the number of common genes between scRNA-seq and scATAC-seq data, and $K$ is the number of cell types in the scRNA-seq annotation. The classification head $g(\cdot)$ is a one-layer FCN with dimension 64-K. In all experiments, we fixed the warm-up epochs $E_1 = 1$, the training epochs $E_2 = 19$, the prototype momentum $\eta = 0.9$, and the reliable data selection threshold $\alpha = 0.95$. To speed up the training, we early stopped the network optimization when the cell classification accuracy of the annotated data reached 99%. We adopted the Gaussian Mixture Model provided in the Scikit-learn Python package[42], v.1.1.1. The scBridge model is trained with mini-batches of size 512 by the AdamW optimizer[43,44] with a learning rate $5e - 4$. All experiments are conducted on an Nvidia RTX 3090 GPU with CUDA 11.4, on the Ubuntu 20.04 OS.

## Data preprocessing

scBridge accepts the gene expression matrix of scRNA-seq data and the gene activity matrix of scATAC-seq data as the inputs. For scRNA-seq data, we first normalized each cell by dividing its total number of read counts on all genes, and then multiplied them by 10,000 to ensure that total counts are the same across cells. After that, we log normalized the read counts and scaled the data to have unit variance and zero means. For scATAC-seq data, we first applied the TF-IDF transformation on the gene activity matrix (more discussions are provided in Supplementary Note 3), and then scaled the data to have unit variance and zero means as well. The detailed preprocessing steps for each dataset are elaborated below:

- **Mouse SNARE-seq cortex data.** The gene expression, gene activity, and peak-by-cell matrices were downloaded from NCBI GEO accession number GSE126074[18]. The fastq downloaded from SRP183521[18] was aligned with the mouse reference genome GRCm38 using bwa[45], v0.7.17-r1198-dirty19, with the parameter of bwa mem. Next, the fragment files were derived from the alignment using sinto, v0.9.0, with the parameters into fragments. In addition, the gene activity matrix was generated using Signac[46], v1.8.0, resulting in a dataset with 9,134 cells and 16,750 genes for the integrative analysis. The cell types were manually annotated according to reported cell-type markers[18,47,48].
- **Human SHARE-seq BMMC data.** The gene expression matrix, peak-by-cell matrix, fragments file, and cell-type annotations were downloaded from NCBI GEO accession number GSE207308[30]. The gene activity matrix was generated using Signac[46], v1.8.0, resulting in a dataset with 78,520 cells and 17,701 genes for the integrative analysis.
- **Mouse 10x Multiome kidney data.** The gene expression, peak-by-cell matrix, and fragments file were downloaded from https://www.10xgenomics.com/resources/datasets/mouse-kidney-nuclei-isolated-with-chromium-nuclei-isolation-kit-saltyez-protocol-and-10x-complex-tissue-dp-ct-sorted-and-ct-unsorted-1-standard[31]. The gene activity matrix was generated using Signac[46], v1.8.0, resulting in a dataset with 14,527 cells and 20,105 genes. The cell types were manually annotated according to the reported cell-type markers[32].
- **Mouse atlas data.** The microfluidic-droplet and FACS raw gene expression count matrices with cell-type annotations of the Mouse atlas scRNA-seq data[32] were downloaded from https://tabula-muris.ds.czbiohub.org/. The unnormalized gene activity matrix and cell-type annotations of the mouse atlas scATAC-seq data[33] were downloaded from https://atlas.gs.washington.edu/mouse-atac/. The peak-by-cell matrix was downloaded from NCBI GEO accession number GSE111586[33]. We renamed some cell types to keep them consistent between scRNA-seq and scATAC-seq data (e.g., "Immature B cell", "Activated B cell", and "B cell" into "B cell"). We reserved cells of 18 common types and 12,689 common genes for the analysis, resulting in a FACS dataset with 21,197 cells, a droplet dataset with 24,965 cells, and a scATAC-seq dataset with 55,941 cells. The FACS and droplet datasets were combined as the scRNA-seq data.
- **Human myocardial infarction data.** The gene expression, gene activity matrix, and clustering results of human myocardial infarction data[34] were downloaded from https://cellxgene.cziscience.com/collections/8191c283-0816-424b-9b61-c3e1d6258a77, with 191,795 snRNA-seq cells and 46,086 scATAC-seq cells. The peak-by-cell matrix was downloaded from https://zenodo.org/record/6578553 and https://zenodo.org/record/6578617. To construct a relatively balanced snRNA-seq dataset, we subsampled $\max\{0.05 n_i, 10,000\}$ cells for cell type $i$ with the number of cells $n_i > 10,000$. All cells are included for cell types with fewer than 10,000 cells, resulting in 67,360 cells from the snRNA-seq data. The number of common genes between two omics is 17,878.
- **Human hematopoiesis data.** The gene expression, gene activity, peak-by-cell matrices, and clustering results of human hematopoiesis data[49] from healthy donors were downloaded

from https://github.com/GreenleafLab/MPAL-Single-Cell-2019. We excluded cells annotated as "Unknown", resulting in 34,609 cells for scRNA-seq data and 33,819 cells for scATAC-seq data for the analysis. The number of common genes between two omics is 15,715.

- **Human peripheral blood mononuclear cells (PBMC)**. The original CITE-seq data, ASAP-seq data, and fragments file are provided in GSE156478[38]. For convenience, we downloaded the preprocessed data provided in https://github.com/SydneyBioX/scJoint/blob/main/data.zip, which contains 4,644 CITE-seq and 4,506 ASAP-seq cells of 7 common types. The peak-by-cell matrix was calculated using the R package ArchR[50], v1.0.2, with default parameters. Though both CITE-seq and ASAP-seq could profile chromatin and protein levels simultaneously, only gene expression and activity matrices are used to focus on the scRNA-seq and scATAC-seq data integration. Additionally, we show that scBridge could also integrate the protein data in Supplementary Note 1.

### Performance and benchmarking

**Baseline methods.** Six single-cell data integration methods were benchmarked for comparisons, including scJoint[15], Seurat[22], Portal[25], Harmony[14], GLUE[24], and Conos[23].

For scJoint[15], we adopted the official code released in https://github.com/SydneyBioX/scJoint, with hyper-parameters provided in the code example or recommended in the paper. Since scJoint first binarizes the gene expression and activity matrices, no data preprocessing is needed. The algorithm directly outputs the embedding of both scRNA-seq and scATAC-seq data (_embeddings.txt), the transferred label for scATAC-seq data (_knn_predictions.txt), as well as the confidence score (_knn_probs.txt).

For Seurat, we used Seurat R package[22], v4.1.4. The raw count matrix of scRNA-seq and the unnormalized gene activity matrix of scATAC-seq were transformed into Seurat objects using the CreateSeuratObject function. With the NormalizeData, FindVariableFeatures, ScaleData, and RunPCA functions, the Seurat objects were further processed, where the PCA dimensions were set to 1:30 for scRNA-seq data and 2:30 for scATAC-seq data, respectively. After that, the FindTransferAnchors function was adopted to identify the anchors between scRNA-seq and scATAC-seq datasets. Then, the TransferData function was employed to impute and integrate the scATAC-seq data, and transfer the annotations from the scRNA-seq dataset into the scATAC-seq cells. The TransferData function also outputs the confidence score of each prediction. Finally, PCA was performed on the combined matrix of scRNA-seq data and imputed scATAC-seq data to obtain features. If not mentioned, all parameters are set as default.

For Portal, we used the portal-sc Python package[25], v1.0.2. Following its default pipeline, we used the model.preprocess function to preprocess the gene expression matrix and gene activity matrix. After that, the model.train function was used to integrate data. We set *training_steps* = 1000 for datasets with sample size <20,000 and *training_steps* = 2000 otherwise, as suggested in the package. Finally, the integration results were obtained from model.latent after running the model.eval function. As Portal itself does not support label transfer, we adopted the KNeighborsClassifier function with $k = 10$ provided in the scikit-learn Python package[42], v1.1.3, to transfer cell annotations from scRNA-seq to scATAC-seq data.

For Harmony, we adopted the same data preprocessing pipeline as scBridge, followed by a PCA dimensional reduction with 50 components, and then used the official harmonypy Python package[14] (https://github.com/slowkow/harmonypy), v0.0.5, with the recommended parameters for data integration. As Harmony itself does not support label transfer, we adopted the KNeighborsClassifier function

with $k = 10$ provided in the scikit-learn Python package[42], v1.1.3, to transfer cell annotations from scRNA-seq to scATAC-seq data.

For GLUE, we adopted the official code released in https://github.com/gao-lab/GLUE, v0.3.2. Following its default pipeline, we first preprocessed the scRNA-seq expression and scATAC-seq peak-by-cell matrices and then constructed the prior regulatory graph. After that, we trained a glue model to integrate data with the scglue.models.fit_SCGLUE function. Then, we applied the model for cell and feature embedding using the encode_data function. Finally, cell annotations with confidence scores were transferred from scRNA-seq to scATAC-seq data with the scglue.data.transfer_labels function.

For Conos, we used conos R package[23], v1.4.6. The raw count matrix of scRNA-seq and the unnormalized gene activity matrix of scATAC-seq were first preprocessed by the basicP2proc function provided in pagoda2 R package[51], v1.0.10. Then, the joint graph was built by the bulidGraph function with parameters $k = 15$, $k.self = 5$, $k.self.weigh = 0.01$, $ncomps = 30$, $n.odgenes = 5e3$, and $space = 'PCA'$. The joint embedding was generated by the embedGraph function. Finally, labels were propagated from the scRNA-seq to the scATAC-seq based on the joint graph using the propagateLabels function, which also outputs the confidence score of each prediction. All parameters are set as default if not mentioned.

**Evaluation metrics.** We adopted the cell classification accuracy (ACC) and weighted F1-score (F1) to measure the performance of label transfer. Specifically, let $\hat{y}_i$ and $y_i$ be the predicted type and the ground-truth annotation for cell $i$, ACC measures the percentage of cells being correctly classified, i.e.,

$$\text{ACC} = \frac{\sum_{i=1}^{N} \delta(\hat{y}_i,(y_i))}{N}, \delta(a,b) = \begin{cases} 1 & \text{if } a = b, \\ 0 & \text{otherwise.} \end{cases} \quad (13)$$

Considering the severe data imbalance among different cell types, ACC would be dominated by large cell types, leading to partial evaluation for those small classes. Hence, we chose the weighted average F1-score as another measurement for comprehensive evaluations, where the weight is inversely proportional to the number of cells for each cell type. Mathematically,

$$\text{F1} = \sum_{k=1}^{K} w_k \frac{2(P_k \cdot R_k)}{P_k + R_k}, w_k = \frac{1/|Y_{y_i}|}{1/|Y_{y_1}| + \cdots + 1/|Y_{y_k}|}, \quad (14)$$

where $P_k, R_k$ denote the Precision and Recall for the $k$-th cell type, and $|Y_{y_i}|$ refers to the number of cells belonging to type $k$.

Silhouette coefficients were adopted to evaluate the joint embedding after data integration. We computed silhouette coefficients given cell types and omics respectively, denoted by $\text{Sil}_{\text{Type}}$ and $\text{Sil}_{\text{Omic}}$. Since one expects the cross-omics cells of the same type to be mixed, higher $\text{Sil}_{\text{Type}}$ and lower $\text{Sil}_{\text{Omic}}$ indicate better results. To provide an overall measure, we harmonized the two silhouette coefficients via

$$\text{Sil}_{\text{F1}} = 2 * \frac{\left[1 - (\text{Sil}_{\text{Omic}} + 1)/2\right]\left[\left(\text{Sil}_{\text{Type}} + 1\right)/2\right]}{\left[1 - (\text{Sil}_{\text{Omic}} + 1)/2\right] + \left[\left(\text{Sil}_{\text{Type}} + 1\right)/2\right]}, \quad (15)$$

where a higher $\text{Sil}_{\text{F1}}$ score indicates better integration results.

To evaluate the performance of novel type prediction performance, we treated it as a novel/seen binary classification task and evaluated the performance via

$$\text{Novel}_{\text{F1}} = \frac{2\text{TP}}{2\text{TP} + \text{FP} + \text{FN}}, \quad (16)$$

where TP, FP, FN stands for true positive, false positive, and false negative, respectively. A higher $\text{Novel}_{F1}$ score corresponds to a more accurate novel-type prediction.

**Trajectory analysis.** The monocle R package[39], v2.22.0, was used to infer the pseudotime in Naive CD4+ T cells and Effector CD4+ T cells from the PBMC dataset, with its default parameters. The inferred pseudotime was then projected to the UMAP plots obtained by scBridge and scJoint.

**Visualization.** We used features extracted by the deep embedding network $f(\cdot)$ as cell representations and adopted the harmony and umap functions provided in the Scanpy Python package[52], v1.9.1, with the default parameters to reduce the dimension to two and visualize cells. Dot plots and feature plots were performed using the ggplot2 R package[53], v3.3.2. Other plots were based on the seaborn Python package[54], v0.11.2.

**Statistical and reproducibility.** Statistical analyses were performed by the SciPy Python package[55], v1.11.1. $p$-value was determined by a two-sided $t$-test, and $p$-value < 0.05 is considered statistically significant. All experiments are conducted under five randomizations with different model initializations. No data were excluded and no statistical methods were used to predetermine sample size. Investigators were not blinded to allocation during library preparation, experiments, or analysis.

### Reporting summary
Further information on research design is available in the Nature Portfolio Reporting Summary linked to this article.

## Data availability
All datasets used in this work are publicly available. Mouse SNARE-seq cortex data[18] used in this study are available in the GEO database with accession ID GSE126074. Human SHARE-seq BMMC data[30] used in this study are available in the GEO database with accession ID GSE207308. Mouse 10x Multiome kidney data[31] used in this study could be downloaded from https://www.10xgenomics.com/resources/datasets/mouse-kidney-nuclei-isolated-with-chromium-nuclei-isolation-kit-saltyez-protocol-and-10x-complex-tissue-dp-ct-sorted-and-ct-unsorted-1-standard. Mouse atlas scRNA-seq data[32] used in this study are available in Figshare (https://figshare.com/projects/Tabula_Muris_Transcriptomic_characterization_of_20_organs_and_tissues_from_Mus_musculus_at_single_cell_resolution/27733). Mouse atlas scATAC-seq data[33] used in this study are available in the GEO database with accession ID GSE111586 and https://atlas.gs.washington.edu/mouse-atac/. Human myocardial infarction data[34] used in this study are available in *Zenodo* databases with accession ID 6578553, 6578617, and https://cellxgene.cziscience.com/collections/8191c283-0816-424b-9b61-c3e1d6258a77. Human hematopoiesis data[49] could be downloaded from https://github.com/GreenleafLab/MPAL-Single-Cell-2019. Human PBMC data[38] could be downloaded from https://github.com/SydneyBioX/scJoint/blob/main/data.zip. Source data are provided in this paper.

## Code availability
The implementation of scBridge is available on https://github.com/XLearning-SCU/scBridge[56].

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

## Acknowledgements

This work was supported in part by the following grants: National Key R&D Program of China under Grant No. 2020YFB1406702 (P), Joint Funds of the National Natural Science Foundation of China under Grant No. U21B2040 (P), Sichuan Science and Technology Program 2021YFS0027 (C), Sichuan Science and Technology Program 2021YFS0403 (C), Joint Funds of the National Natural Science Foundation of China under Grant No. 62125201 (Y) and No. 62020106007 (Y).

## Author contributions

X.P. and Y.L. conceived the study. Y.L. and M.Y. designed and implemented the scBridge algorithm. Y.L., D.Z., M.Y., D.P, J.Y., and J.L. evaluated scBridge and other baselines on seven datasets. D.Z. and L.C. preprocessed the data and analyzed the results. Y.L. participated in the revision and evaluated baselines. All authors participated in writing the manuscript.

## Competing interests

The authors declare no competing interests.
