## [Peer Review File · Nature Communications]

scBridge embraces cell heterogeneity in single-cell RNA-seq and ATAC-seq data integrationREVIEWER COMMENTS

Reviewer #1 (Remarks to the Author):

In this article, Li et al. develop the method scBridge to integrate scRNA-seq and scATAC-seq. scBridge requires the input of well-annotated scRNA-seq data and uses neural networks to train lower-dimension embeddings of scRNA-seq. By inputting scATAC-seq, it will then adjust the embedding function for a few rounds to narrow the gap between the two modalities. The researchers have shown that scBridge outperforms other methods for data integration and cell-type label transfer. They also provide evidence to prove the usefulness of scBridge under the condition of inconsistent cell type across two modalities, different amounts of scRNA-seq data, and different dropout rates in either modality. The paper is well-structured with nicely generated graphs. I also have some comments:

Major comments:

1. In your paper, I noticed that you have mentioned in lines 246-247 that “we observe that a portion of scATAC-seq cells exhibit smaller omics differences with scRNA-seq cells when their chromatin accessibility contributes more to transcription”. Could you show me any evidence of those cells? Or give any reference to the saying?
2. In figure 1, I would highly recommend adding the structure of the deep neural classifier which is f and g. This can help other researchers understand the overall design of the method.
3. I have noticed that most of your results use tSNE as visualization. Figure 6 uses UMAP. I would recommend using UMAP for all figures.
4. Figure 2 is the essential part of the paper. A single dataset from SNARE-seq is not enough to make a solid conclusion. I would recommend using datasets from SNARE-seq, SHARE-seq, and 10X Multiome datasets. This experiment should also include at least 3 tissue types. You should also get the comparison result with different down sample rates with those datasets.
5. I recommend speed comparison across different methods. This can be done using the single-cell multiomics datasets mentioned above.
6. In figure 4b-c, how do you compute the confidence score for scBridge and Seurat? How about other methods such as scJoint? I recommend calculating the accuracy of predicting novel cell types in all methods.

Minor comments:

1. Figure 2A should also be colored by different modalities. Lines 153-154 claim setting up the confidence score using distribution. Could you include the distribution and your cutoff in your supplementary figure?
2. In lines 162-163, the majority of dropout events are caused by lower sequencing depth. Therefore, the way to simulate the dropout events can be wrong. I would recommend performing down sampling from the reads level and re-run the experiment.
3. Line 316-317, I noticed you applied the TF-IDF to the gene activity matrix. Do you have any references to do so? How do you scale the data to follow normal distribution?

Reviewer #2 (Remarks to the Author):

In this manuscript, Li and colleagues proposed an algorithm, scBridge, for integrating scRNA-seq data and scATAC-seq data. The proposed method accounts for varying cross-omics differences in different cell populations. The authors applied scBridge to five datasets, which showed the reliability of their proposed method.

Q1. Would that be possible to benchmark with GLUE and Portal, which are also deep learning-based methods that can integrate scRNA-seq and scATAC-seq datasets?

Cao, Z. J., & Gao, G. (2022). Multi-omics single-cell data integration and regulatory inference with graph-linked embedding. *Nature Biotechnology*, 40(10), 1458-1466.

Zhao, J., Wang, G., Ming, J., Lin, Z., Wang, Y., Wu, A. R., & Yang, C. (2022). Adversarial domain translation networks for integrating large-scale atlas-level single-cell datasets. *Nature Computational Science*, 2(5), 317-330.

Q2. The authors mentioned that there is an optional loss in scBridge named structure preservation loss, which can handle the case that there are some scATAC-seq-unique cell types. They applied this loss on the integration of human myocardial infarction data. But in general, without scATAC-seq annotations, it is hard to anticipate such a situation. The authors may need to provide users with more clear guidance about when to use the structure preservation loss.

Q3. The authors may need to compare the computation efficiency of scBridge with other methods.

The major updates we made in the revised version are summarized as follows:

- **[Datasets]** In addition to the SNARE-seq dataset of mouse brain cortex¹, we performed new experiments on another two golden benchmarks, namely, the SHARE-seq dataset of human bone marrow² and the 10x Multiome dataset of mouse kidney³ (page 4, Fig. 2 in manuscript). Moreover, we also carried out down-sampling experiments on the three datasets. Now we have seven datasets in total for evaluation.
- **[Baselines]** In addition to scJoint⁴, Seurat⁵, Harmony⁶, and Conos⁷, we further compared scBridge with two deep learning-based methods GLUE⁸ and Portal⁹. Extensive results demonstrate the superiority of scBridge over six baselines in data integration and label transfer performance.
- **[Novel Type Discovery]** We supplied details of how confidence scores are obtained by scBridge and baselines (page 15, section *Baseline methods* in manuscript). Moreover, we designed a data-driven strategy to estimate the confidence threshold for distinguishing seen and novel types (page 8 in manuscript), as well as a new metric to evaluate the novel type discovery performance (page 16, section *Evaluation metrics* in manuscript). Finally, we provided guidance on when to use the structure preservation loss for novel type discovery (Supplementary Note 2).
- **[Computational Efficiency]** We newly evaluated the running time and memory consumption of all tested methods on mouse atlas subsets of 5,000–80,000 cells (page 3, Fig. 3c in manuscript). The results show that scBridge requires linearly increasing time (the third-best) and constant memory consumption (the second-best) with respect to cell numbers, which is favorable for handling large-scale data.

In addition to the above major updates, according to the review comments, we also updated the expressions about the variety of omics differences (page 2, section *Introduction* in manuscript), added detailed network structures of scBridge (page 3, Fig. 1 in manuscript), reconducted the dropout experiments (page 9, section *scBridge is robust to the dropout technical noise in sequencing data* in manuscript), unified all figures with UMAP visualization, and presented the discussion on the TF-IDF preprocessing for scATAC-seq data (Supplementary Note 3). The point-by-point response to the reviewers' comments is provided below, with the corresponding revised manuscript attached at the end of each response. To aid in clarity, we highlighted the revisions in color.

Point-by-point Response to Reviewer #1

[Overall Comment] In this article, Li et al. develop the method scBridge to integrate scRNA-seq and scATAC-seq. scBridge requires the input of well-annotated scRNA-seq data and uses neural networks to train lower-dimension embeddings of scRNA-seq. By inputting scATAC-seq, it will then adjust the embedding function for a few rounds to narrow the gap between the two modalities. The researchers have shown that scBridge outperforms other methods for data integration and cell-type label transfer. They also provide evidence to prove the usefulness of scBridge under the condition of inconsistent cell type across two modalities, different amounts of scRNA-seq data, and different dropout rates in either modality. The paper is well-structured with nicely generated graphs.

[Response] Thanks for the positive comments on our work.

[Point #1] In your paper, I noticed that you have mentioned in lines 246-247 that “we observe that a portion of scATAC-seq cells exhibit smaller omics differences with scRNA-seq cells when their chromatin accessibility contributes more to transcription”. Could you show me any evidence of those cells? Or give any reference to the saying?

[Response #1] We apologize that the previous expression is not precise enough. In the revision, we updated the words as “we observe that a portion of scATAC-seq cells exhibit smaller omics differences with scRNA-seq cells when their chromatin accessibility has higher positive correlations with gene expression”. To be specific, as a reflection of cell heterogeneity, the previous works^{10,11} reported that the chromatin accessibility of scATAC-seq cells exhibits various correlations with gene expression of scRNA-seq. In our experiments, as shown in Supplementary Fig. 1a, the correlation between chromatin accessibility and gene expression decreases as the iteration of scBridge proceeds. By taking Ex-L2/3-Rasgrf2 cells as an example, Supplementary Fig. 1b shows a decrease in the *Rasgrf2* gene activity as the model iterates. Meanwhile, as shown in

Supplementary Fig. 1c, the discrepancy between *Rasgrf2* gene activity and gene expression increases as integration proceeds, consistent with the results in Supplementary Fig. 1a. Correspondingly, scBridge achieves higher label transfer accuracy for scATAC-seq cells selected in the early iterations since they exhibit smaller omics differences with scRNA-seq cells, as illustrated in Fig. 1c. These experimental results and previous works^{10,11} support our claim that “we observe that a portion of scATAC-seq cells exhibit smaller omics differences with scRNA-seq cells when their chromatin accessibility has higher positive correlations with gene expression”.

According to your comments, we updated the words in the *Introduction*, *Results*, and *Method* sections as follows.

Introduction

Here, we reveal that instead of being an interference, the cell heterogeneity could be exploited to facilitate data integration based on the following observation. **Specifically, the chromatin accessibility of scATAC-seq cells exhibits variable correlations with gene expression of scRNA-seq^{10,11}. scATAC-seq cells with higher positive correlation exhibit smaller omics differences, which hence are easier to be integrated and could bridge the modality gap between the two omics.** According to the observation, we design scBridge, a heterogeneous transfer learning method for multi-omics data integration. Briefly, scBridge first warms up a deep neural classifier with the annotated scRNA-seq data, and then identifies the scATAC-seq cells with smaller omics differences through reliability modeling. After that, the reliable scATAC-seq cells are integrated with scRNA-seq cells through cross-omics prototype alignment. Lastly, scBridge selects and merges the most reliable scATAC-seq cells into the annotated scRNA-seq data to narrow the omics gap. By repeating the above processes, the omics difference would be gradually reduced, and more cells would be integrated, leading to the final integration result.

Results

scBridge achieves promising integration results on the golden benchmarks.

To intuitively show how scBridge iteratively integrates scRNA-seq and scATAC-seq data through heterogeneous transfer learning, we visualized the integration process on the SNARE-seq dataset in Fig. 2c. To be specific, the right figure shows the Pearson correlation score (computed on all genes) between scRNA-seq cells and the selected reliable scATAC-seq cells, where larger scores denote smaller omics differences between scRNA-seq and scATAC-seq cells. As shown, scBridge first integrates scATAC-seq cells that are most similar to scRNA-seq cells, and gradually integrates more distinct ones in the subsequent iterations (T test p value $< 1e-3$ in the first five iterations, with the Pearson correlation score decreasing significantly). Such a trend also holds in different types of cells as illustrated in Supplementary Fig. 1a. **Here, we took Ex-L2/3-Rasgrf2 cells of scATAC-seq as an example to demonstrate various cell correlation levels across omics.** Supplementary Fig. 1b shows a decrease in the *Rasgrf2* gene activity as the model iterates. Meanwhile, Supplementary Fig. 1c demonstrates that the discrepancy between *Rasgrf2* gene activity and gene expression increases as integration proceeds, consistent with the results in Supplementary Fig. 1a. The left figure in Fig. 2c demonstrates the reliable cell selection and overall label transfer accuracy across iterations. In brief, after the first iteration, scBridge achieves 60.11% label transfer accuracy for all scATAC-seq cells. Based on the Gaussian Mixture Model, 1,898 scATAC-seq cells are selected as the annotated data with an accuracy of 90.89%. By using those reliable scATAC-seq cells to bridge RNA and ATAC omics, scBridge achieves better integration results (63.62% label transfer accuracy) in the second iteration. As the training proceeds, more scATAC-seq cells are selected as reliable by scBridge, and the label transfer accuracy steadily grows to 71.95%.

Methods

Reliability modeling for scATAC-seq data with the Gaussian Mixture.

After the warm-up, the model is able to correctly predict cell types for scRNA-seq data. However, the classification performance on scATAC-seq data is undesirable due to the modality gap between RNA and ATAC omics. Nevertheless, thanks to cell heterogeneity, we observe that a portion of scATAC-seq cells exhibit smaller omics differences with scRNA-seq cells **when their chromatin accessibility has higher positive correlations with gene expression**, and those cells are easier to be integrated. Specifically, the deep embedding network $f(\cdot)$ tends to extract more discriminative features, and the classifier $g(\cdot)$ tends to make more confident predictions for those cells. Motivated by such an observation, to identify those reliable cells that have smaller omics differences, we model the reliability of scATAC-seq cells with Gaussian Mixture based on their discriminability and confidence.

Supplementary Figure 1. a, The Pearson correlation score between scRNA-seq and the selected scATAC-seq cells of 20 classes in different iterations on the SNARE-seq dataset. Specifically, we first averaged the normalized gene expression of scRNA-seq cells for each type. Then, we computed the Pearson correlation score between the normalized gene activity of each scATAC-seq cell and the mean gene expression according to the predicted cell type. **Each boxplot ranges from the upper and lower quartiles with the median as the horizontal line and whiskers extend to 1.5 times the interquartile range.** **b**, Iteration and the gene activity of marker gene *Rasgrf2*¹ projected on the UMAP plot of Ex-L2/3-Rasgrf2 cells of scATAC-seq. **c**, Dynamic change of *Rasgrf2* gene expression and gene activity in Ex-L2/3-Rasgrf2 cells of scRNA-seq and scATAC-seq, respectively. The gray area denotes the standard error.

Editorial Note: Parts of panel (a) in Figure 1 below have been redacted as indicated to remove third-party material where no permission to publish could be obtained.

[Point #2] In figure 1, I would highly recommend adding the structure of the deep neural classifier which is *f* and *g*. This can help other researchers understand the overall design of the method.

[Response #2] Thanks for the valuable suggestion. To provide a more comprehensive overview of scBridge, we added the deep neural encoder *f* and classifier *g* in Fig. 1 and revised its caption as suggested.

[Point #3] I have noticed that most of your results use tSNE as visualization. Figure 6 uses UMAP. I would recommend using UMAP for all figures.

[Response #3] According to your suggestion, we have now unified and updated all figures in the manuscript and supplementary material with UMAP visualization.

[Points #4 & #7] Figure 2 is the essential part of the paper. A single dataset from SNARE-seq is not enough to make a solid conclusion. I would recommend using datasets from SNARE-seq, SHARE-seq, and 10x Multiome datasets. This experiment should also include at least 3 tissue types. You should also get the comparison result with different down sample rates with those datasets. Figure 2A should also be colored by different modalities.

[Responses #4 & #7] Thank you for the constructive comments. In addition to the SNARE-seq dataset of mouse brain cortex¹, we have now included the SHARE-seq dataset of human bone marrow² and the 10x Multiome dataset of mouse kidney³. Both the standard and down-sampling experiments on the three datasets are also conducted and the results are shown in Fig. 2 and Supplementary Fig. 2–4. In short, the results demonstrate that our scBridge achieves the best performance on all three datasets under different down-sampling rates.

Due to the space limitation, we found it is hard to place more figures in Fig. 2. Hence, we presented the UMAP visualizations colored by different modalities on the SNARE-seq data in Supplementary Fig. 2a. Moreover, we newly attached the UMAP visualizations of scBridge and scJoint on the SHARE-seq and 10x Multiome in Fig. 2e and f, colored by both cell types and modalities.

For your convenience, we attached the major revisions below.

Figure 2. Integration results on three golden benchmarks. **a**, UMAP plots of the joint embeddings obtained by the seven methods on the SNARE-seq dataset, where cells are colored by types. **b**, Quantitative evaluation on the SNARE-seq dataset in terms of the joint embedding quality and label transfer accuracy. **c**, (Left) The number of reliable scATAC-seq cells selected by scBridge with the corresponding accuracy, and the overall label transfer accuracy across the training process on the SNARE-seq dataset. (Right) The Pearson correlation score between scRNA-seq and the selected scATAC-seq cells in different iterations. **d**, The label transfer matrix of the agreement between the predicted cell type and the ground-truth annotation. A clearer diagonal structure denotes better label transfer performance. **e**, UMAP embeddings of scBridge and scJoint on the SHARE-seq dataset. **f**, UMAP embeddings of scBridge and scJoint on the 10x Multiome dataset. The first and second rows show cells colored by types and omics, respectively. **g**, The label transfer accuracy and F1-score of the tested methods on three benchmarks, where 100%, 75%, 50%, and 25% annotated scRNA-seq data are used. Five random experiments are conducted with different downsample rates. Each boxplot ranges from the upper and lower quartiles with the median as the horizontal line and whiskers extend to 1.5 times the interquartile range.

scBridge achieves promising integration results on the golden benchmarks.

To evaluate the integration performance of scBridge, we first applied it to three golden benchmarks including the SNARE-seq dataset of mouse brain cortex¹, the SHARE-seq dataset of human bone marrow², and the 10x Multiome dataset of mouse kidney³. As these three sequencing techniques could link the cell's transcriptome with its accessible chromatin, the pairing information provides a golden criterion to validate the integration performance. Notably, the pairing information was not used during integration, but only for validation. Moreover, these three datasets cover three different tissues and two species, which also evaluates the generalization ability of the methods.

To intuitively show how scBridge iteratively integrates scRNA-seq and scATAC-seq data through heterogeneous transfer learning, we visualized the integration process on the SNARE-seq dataset in Fig. 2c. To be specific, the right figure shows the Pearson correlation score (computed on all genes) between scRNA-seq cells and the selected reliable scATAC-seq cells, where larger scores denote smaller omics differences between scRNA-seq and scATAC-seq cells. As shown, scBridge first integrates scATAC-seq cells that are most similar to scRNA-seq cells, and gradually integrates more distinct ones in the subsequent iterations (T test p value $< 1e-3$ in the first five iterations, with the Pearson correlation score decreasing significantly). Such a trend also holds in different types of cells as illustrated in Supplementary Fig. 1a. Here, we took Ex-L2/3-Rasgrf2 cells of scATAC-seq as an example to demonstrate various cell correlation levels across omics. Supplementary Fig. 1b shows a decrease in the *Rasgrf2* gene activity as the model iterates. Meanwhile, Supplementary Fig. 1c demonstrates that the discrepancy between *Rasgrf2* gene activity and gene expression increases as integration proceeds, consistent with the results in Supplementary Fig. 1a. The left figure in Fig. 2c demonstrates the reliable cell selection and overall label transfer accuracy across iterations. In brief, after the first iteration, scBridge achieves 60.11% label transfer accuracy for all scATAC-seq cells. Based on the Gaussian Mixture Model, 1,898 scATAC-seq cells are selected as the annotated data with an accuracy of 90.89%. By using those reliable scATAC-seq cells to bridge RNA and ATAC omics, scBridge achieves better integration results (63.62% label transfer accuracy) in the second iteration. As the training proceeds, more scATAC-seq cells are selected as reliable by scBridge, and the label transfer accuracy steadily grows to 71.95%.

Fig. 2a and Supplementary Fig. 2a illustrate the final data integration results achieved by scBridge and six baseline methods. As shown, though all seven methods successfully mix scRNA-seq and scATAC-seq cells, scBridge and scJoint achieve more discriminative cell clusters compared with other baselines. In some clusters, however, scJoint falsely integrates cells with different types, leading to inferior label transfer performance. To further validate the superiority of scBridge, Fig. 2d and Supplementary Fig. 2b visualize the confusion matrix of the transferred labels. The results show that scBridge discriminates the cells of different types more accurately compared with all baselines. For example, scJoint fails to separate Claustrium, Mic, and OPC cells, whereas scBridge achieves almost perfect label transfer on them. By using the silhouette score and label transfer accuracy to quantitatively evaluate the integration results, Fig. 2b shows that scBridge achieves the highest harmonized silhouette score, indicating its superiority in the removal of omics difference and the preservation of cell type difference. We also noticed that scBridge achieves a more precise integration for the rare cell types, *i.e.*, a significant improvement on the weighted F1-score in label transfer (42.26% by scBridge compared with 22.12% by scJoint).

We further visualized the joint embeddings obtained by scBridge and scJoint on the SHARE-seq and 10x Multiome datasets in Fig. 2e–f. On the SHARE-seq dataset, scBridge achieves better cell grouping than scJoint, especially for the rare types like Baso. On the 10x Multiome dataset, scBridge successfully mixes scRNA-seq and scATAC-seq cells, while scJoint fails to eliminate the gap between the two modalities. The UMAP visualizations, label transfer matrix, and quantitative metrics of scBridge and all other baselines in Supplementary Fig. 3–4 demonstrate the superior performance

of scBridge in data integration and label transfer.

Finally, as the heterogeneous transfer learning paradigm of scBridge requires the annotated scRNA-seq data, a natural question is how many annotated scRNA-seq cells are needed for accurate integration. To answer this question, we evaluated the robustness of scBridge against the number of annotations on the three golden benchmarks, compared with four baselines that support label transfer. Specifically, we carried out experiments by using 100%, 75%, 50%, and 25% of scRNA-seq data. Fig. 2g shows that scBridge achieves the best label transfer accuracy and F1 score under all downsample rates on three benchmarks. Notably, on the 10x Multiome dataset, scBridge remains a high average F1 score of 77.08% with only 25% annotated scRNA-seq cells compared with 77.35% on full data. In contrast, scJoint encounters a significant performance drop in average F1 score, *i.e.*, from 73.22% on full data to 59.36% on 25% downsampled data (T test p value = $1.94e-5$). In addition, scBridge with only 50% scRNA-seq annotations outperforms all baselines with full data on the SNARE-seq dataset. Such a data-efficient property of scBridge could attribute to its heterogeneous transfer learning paradigm. Namely, as long as the annotated scRNA-seq data is enough for identifying a portion of reliable scATAC-seq data, scBridge could progressively integrate the rest cells.

[Point #5] I recommend speed comparison across different methods. This can be done using the single-cell multi-omics datasets mentioned above.

[Response #5] Thanks for the suggestion. In the revision, we evaluated the running time and memory consumption of scBridge on mouse atlas subsets of 5,000–80,000 cells compared with six baselines. The results in Fig. 2c show that scBridge requires linearly increasing time with respect to cell numbers, and achieves the third-best time efficiency. Moreover, scBridge requires a constant memory consumption for the arbitrary number of cells and achieves the second-best memory efficiency. The linear time complexity and constant memory complexity make scBridge a promising tool for handling large-scale data.

For your convenience, we attached the corresponding revisions below.

scBridge scales to atlas data.

With the development of sequencing techniques, the number of cells profiled with various protocols grows continually, arousing the demand for efficiently handling large-scale data. To access how scBridge scales to large data, we evaluated it on the mouse atlas dataset. Specifically, we used the cells sequenced with FACS and droplet protocols provided by Tabula Muris¹² as scRNA-seq data, and the cells sequenced by Cusanovich et al.¹³ as scATAC-seq data. After data preprocessing, 102,103 cells from 18 common types are selected for evaluation.

To investigate the computation efficiency of scBridge, we applied it to five subsets of mouse atlas with 5,000–80,000 cells. Fig. 3c shows the (logged) running time and memory consumption of all tested methods with respect to different cell numbers. As shown, scBridge takes linearly increasing running time (the third-best) and constant memory consumption (the second-best), which is favorable in scaling to large data.

Figure 3. Integration results on mouse atlas data. **a**, UMAP visualization of the joint embeddings learned by scBridge and scJoint. The first and second rows show cells colored by types and omics, respectively. **b**, The cell type and 1–omics silhouette coefficients of scBridge and six baselines. A higher cell type silhouette coefficient indicates better biological grouping, and a higher 1–omics silhouette coefficient indicates better cross-omics cell mixing. Each boxplot ranges from the upper and lower quartiles with the median as the horizontal line and whiskers extend to 1.5 times the interquartile range. **c**, The running time and memory consumption of different methods on mouse atlas subsets of 5,000–80,000 cells. **d**, The dot plot of relative expression of marker genes *Tgfb1*, *Cd68*, *Cd34* in cells predicted as Hematopoietic stem cell (HSC), Macrophages, and Monocytes by scBridge. The size of the circle represents the proportion of expressing cells, and the color indicates the average expression level. **e**, The agreement between the predicted label and the manual annotation. Matrices with a clearer diagonal structure indicate better performance.

[Point #6] In figure 4b-c, how do you compute the confidence score for scBridge and Seurat? How about other methods such as scJoint?

[Response #6] Thanks for pointing out the missing details. The confidence score of scBridge is computed based on the estimated cell reliability introduced in the section of *Reliability modeling for scATAC-seq data with the Gaussian Mixture* (page 13). We added its mathematical definition in the revision as follows.

After training, scBridge computes the confidence score r_i^c for each scATAC-seq cell based on the estimated cell reliability, namely,

$$r_i^c = p(c_2^a | d_i^a) p(c_1^b | l_i^b), \quad (12)$$

where $p(c_2^a | d_i^a) \in [0, 1]$ corresponds to the probability of cell i belonging to the second component c_2^a in GMM g^a , and $p(c_1^b | l_i^b) \in [0, 1]$ the probability belonging to the first component c_1^b in GMM g^b , following the definitions in Eq. 4 and 5. With the structure preservation loss, scBridge would assign a lower confidence score for the scATAC-seq cells with unseen types, thus enabling novel type discovery.

In the revision, in addition to Seurat and scJoint, we computed the confidence of another two baselines GLUE and Conos that themselves support label transfer. The confidence score of all these four baselines are directly computed by the provided functions in their packages. We added the corresponding details in the *Performance and Benchmarking* section as follows.

Performance and benchmarking.

Baseline methods.

Six single-cell data integration methods were benchmarked for comparisons, including scJoint⁴, Seurat⁵, Portal⁹, Harmony⁶, GLUE⁸, and Conos⁷.

For scJoint⁴, we adopted the official code released in <https://github.com/SydneyBioX/scJoint>, with hyper-parameters provided in the code example or recommended in the paper. Since scJoint first binarizes the gene expression and activity matrices, no data preprocessing is needed. The algorithm directly outputs the embedding of both scRNA-seq and scATAC-seq data (`_embeddings.txt`), the transferred label for scATAC-seq data (`_knn_predictions.txt`), as well as the confidence score (`_knn_probs.txt`).

For Seurat, we used Seurat R package⁵, v4.1.4. The raw count matrix of scRNA-seq and the unnormalized gene activity matrix of scATAC-seq were transformed into Seurat objects using the `CreateSeuratObject` function. With the `NormalizeData`, `FindVariableFeatures`, `ScaleData`, and `RunPCA` functions, the Seurat objects were further processed, where the PCA dimensions were set to 1 : 30 for scRNA-seq data and 2 : 30 for scATAC-seq data, respectively. After that, the `FindTransferAnchors` function was adopted to identify the anchors between scRNA-seq and scATAC-seq datasets. Then, the `TransferData` function was employed to impute and integrate the scATAC-seq data, and transfer the annotations from the scRNA-seq dataset into the scATAC-seq cells. The `TransferData` function also outputs the confidence score of each prediction. Finally, PCA was performed on the combined matrix of scRNA-seq data and imputed scATAC-seq data to obtain features. If not mentioned, all parameters are set as default.

For Portal, we used the portal-sc Python package⁹, v1.0.2. Following its default pipeline, we used the `model.preprocess` function to preprocess the gene expression matrix and gene activity matrix. After that, the `model.train` function was used to integrate data. We set `training_steps = 1,000` for datasets with sample size < 20,000 and `training_steps = 2,000` otherwise, as suggested in the package. Finally, the integration results were obtained from `model.latent` after running the `model.eval` function. As Portal itself does not support label transfer, we adopted the `KNeighborsClassifier` function with $k = 10$ provided in the scikit-learn Python package¹⁴, v1.1.3, to transfer cell annotations from scRNA-seq to scATAC-seq data.

For Harmony, we adopted the same data preprocessing pipeline as scBridge, followed by a PCA dimensional reduction with 50 components, and then used the official `harmonypy` Python package⁶ (<https://github.com/slowkow/harmonypy>), v0.0.5, with the recommended parameters for data integration. As Harmony itself does not support label transfer, we adopted the `KNeighborsClassifier` function with $k = 10$ provided in the scikit-learn Python package¹⁴, v1.1.3, to transfer cell annotations from scRNA-seq to scATAC-seq data.

For GLUE, we adopted the official code released in <https://github.com/gao-lab/GLUE>. Following its default pipeline, we first preprocessed the scRNA-seq expression and scATAC-seq peak-by-cell matrices and then constructed

the prior regulatory graph. After that, we trained a glue model to integrate data with the `scglue.models.fit_SCGLUE` function. Then, we applied the model for cell and feature embedding using the `encode_data` function. Finally, cell annotations with confidence scores were transferred from scRNA-seq to scATAC-seq data with the `scglue.data.transfer_labels` function.

For Conos, we used `conos` R package⁷, v1.4.6. The raw count matrix of scRNA-seq and the unnormalized gene activity matrix of scATAC-seq were first preprocessed by the `basicP2proc` function provided in `pagoda2` R package¹⁵, v1.0.10. Then, the joint graph was built by the `bulidGraph` function with parameters $k = 15$, $k.self = 5$, $k.self.weigh = 0.01$, $ncomps = 30$, $n.odgenes = 5e3$, and $space = 'PCA'$. The joint embedding was generated by the `embedGraph` function. Finally, labels were propagated from the scRNA-seq to the scATAC-seq based on the joint graph using the `propagateLabels` function, which also outputs the confidence score of each prediction. All parameters are set as default if not mentioned.

[Point #8] Lines 153-154 claim setting up the confidence score using distribution. Could you include the distribution and your cutoff in your supplementary figure? I recommend calculating the accuracy of predicting novel cell types in all methods.

[Response #8] In the previous version, we manually set the cutoff value by observing the confidence distribution in the violin plot. Motivated by your comments, we realized that a more general and practical solution would be estimating the cutoff value in a data-driven manner. Specifically, we proposed to fit the confidence score of all scATAC-seq cells with two-component GMM, and then estimate the confidence threshold using the intersection of two probability density functions (PDF). The same strategy was adopted for scBridge and baseline methods. The details were supplied in the Supplementary Note 2 and attached below for your convenience.

Here we supplied the detailed pipeline of novel type discovery. After the training with the structure preservation loss L_{STC} , scBridge would compute a confidence score ranged $[0, 1]$ for each of the n' scATAC-seq cells, denoted as $r' = \{r'_1, r'_2, \dots, r'_{n'}\}$. To predict which portion of cells is novel, we derived a confidence threshold in a data-driven manner. Specifically, we fit the distribution of r' with a two-component GMM g , namely,

$$g(r') = \gamma_{c_1} \phi(d^t | c_1) + \gamma_{c_2} \phi(d^t | c_2), \quad (13)$$

where $\gamma_{c_1}, \gamma_{c_2}$ denote the mixture coefficient for components c_1, c_2 . Let $p(c_1 | r'_i), p(c_2 | r'_i) \in [0, 1]$ be the probability of cell i belonging to the two components, the confidence threshold dividing novel and seen cell types is estimated as β that satisfies $p(c_1 | r'_i) = p(c_2 | r'_i)$. If there is more than one suitable $\beta \in [0, 1]$, we chose the smallest one as the confidence threshold. Finally, those scATAC-seq cells with $r'_i < \beta$ are predicted as novel. The confidence score distribution, the two GMM components, and the estimated confidence threshold of scBridge and other baselines are illustrated in Fig. 4b and Supplementary Fig. 6e.

According to your advice, we illustrated the distribution of the confidence score, the GMM components, and the estimated cutoff value for scBridge and baselines. The revised figures and manuscript (page 7) are attached below.

Furthermore, we conducted a more challenging evaluation by manually removing the Myeloid cells from scRNA-seq data. In other words, there are only 7 cell types in common for scRNA-seq and scATAC-seq data, and both of them have unique cell types. The UMAP visualizations in Fig. 4a and Supplementary Fig. 6a illustrate that scJoint fails to integrate the cells from different omics, and other methods achieve less distinct partition of cells with different types compared with scBridge. According to the label transfer matrix in Fig. 4d, scBridge and Seurat transfer fewer scATAC-seq cells of common types to the three unique types in scRNA-seq data than GLUE and Conos, and scBridge achieves more precise label transfer results among the seven common types. Next, we focused on the integration results for Myeloid cells in scATAC-seq data, which is novel with respect to the annotations in scRNA-seq data. Equipped with the structure preservation loss, scBridge assigns a relatively low confidence score for scATAC-seq Myeloid cells as shown in Fig. 4b. To identify cells of novel types, instead of manually setting a confidence threshold, we proposed a data-driven strategy by fitting the confidence score of all cells with a two-component GMM. As shown in Fig. 4b, the confidence threshold is estimated by the intersection of two probability density functions (PDF). In other words, cells belonging to the less confident GMM component are considered novel. According to the novel type identification performance shown in Fig. 4c and Supplementary Fig. S6d, scBridge gives a more distinct pattern between cells of common and novel types, leading to the highest F1 score for novel type discovery. The superiority of scBridge is also verified by the label transfer matrix in Fig. 4d, namely, it assigns fewer cells of common types as novel.

Figure 4. Integration results on the human myocardial infarction data, where both RNA and ATAC omics have their unique cell types (Adipocyte, Cycling cells, and Mast cells only exist in scRNA-seq data, and Myeloid cells only exist in scATAC-seq data). **a**, UMAP visualization of the joint embedding obtained by scBridge, scJoint, Seurat, GLUE, and Conos. The first and second rows show cells colored by types and omics, respectively. The novel Myeloid cells are gray-colored and red-circled. **b**, (Left) scBridge’s UMAP embedding of scATAC-seq cells, colored by the confidence score. (Right) scBridge’s novel type threshold estimated by applying a two-component GMM on the confidence score. **c**, The confidence score predicted by scBridge and Seurat on different types of scATAC-seq cells. **d**, The label transfer results of scBridge, Seurat, GLUE, and Conos. Cells are considered novel if their confidence scores are below the threshold estimated by GMM.

Supplementary Figure 6. **a**, UMAP visualization of Portal and Harmony on the human myocardial infarction data with novel types. The first and second rows show cells colored by types and omics, respectively. **b**, UMAP visualization of scATAC-seq cells obtained by scJoint, Seurat, GLUE, and Conos, where cells are colored by the confidence score. **c**, The label transfer results of scJoint. Cells with a confidence score below the GMM-estimated threshold are considered novel. **d**, The confidence score for different types of scATAC-seq cells predicted by scJoint, GLUE, and Conos. **e**, The novel type threshold of scJoint, Seurat, GLUE, and Conos, which is estimated by applying a two-component GMM on the confidence score. **f**, UMAP visualization of scATAC-seq cells obtained by scBridge, where cells colored by the confidence score on four cases. Namely, from left to right, enabling structure loss on data with novel cell type, disabling structure loss on data with novel cell type, enabling structure loss on data without novel cell type, and disabling structure loss on data without novel cell type.

Finally, we quantitatively evaluated the novel type discovery performance by the F1 score of a novel/seen binary classification task. The results of scBridge and baselines are reported in the violin plot in Fig. 4c and Supplementary Fig. 6d. The definition of the metric (page 16) is attached below.

To evaluate the performance of novel type prediction performance, we treated it as a novel/seen binary classification task and evaluated the performance via

$$\text{Novel}_{F1} = \frac{2TP}{2TP + FP + FN}, \quad (16)$$

where TP,FP,FN stands for true positive, false positive, and false negative, respectively. A higher Novel_{F1} score corresponds to a more accurate novel type prediction.

[Point #9] In lines 162-163, the majority of dropout events are caused by lower sequencing depth. Therefore, the way to simulate the dropout events can be wrong. I would recommend performing down sampling from the reads level and re-run the experiment.

[Response #9] Thanks for your valuable comments. The most direct way to simulate the dropout event is to perform downsampling on the raw reads of scATAC-seq and scRNA-seq. To this end, we resorted to the scATAC-seq fastq and bam files of human hematopoiesis data provided on GEO Accession GSE139369¹⁶. However, barcode files are unavailable in scATAC-seq SRA files and bam files are not publicly accessible, which hinders downstream analyses. Therefore, to simulate read-level dropout events, we performed downsampling on the raw gene activity matrix and gene expression matrix by the `downsampleMatrix` function provided in the `Scuttle R` package¹⁷.

According to your advice, we re-conducted the dropout experiments with the simulated data. Specifically, we compared scBridge with all six baselines on downsampled scRNA-seq data. On downsampled scATAC-seq data, we evaluated scBridge and the other five baselines except GLUE since it takes the peak-by-cell instead of gene activity matrix as the input.

In short, scBridge achieves superior integration performance in dropout experiments. For your convenience, we attached the revised figures and manuscript (page 9) below.

To investigate the robustness of scBridge against the dropout technical noise, we applied it on the human hematopoiesis data which contains 34,609 scRNA-seq and 33,819 scATAC-seq cells from 23 common types. To simulate the dropout events, we downsampled the scRNA-seq count matrix and scATAC-seq activity matrix by 25%, 50%, and 75%, respectively, with the `downsampleMatrix` function provided in the `scuttle R` package¹⁷. As GLUE takes the peak-by-cell instead of gene activity matrix as the input, we evaluated its performance only on RNA dropout data. As shown in Fig. 5c, scBridge achieves superior robustness towards the scRNA-seq data quality. Namely, its integration and label transfer performances are almost impervious under up to 75% dropout rate on scRNA-seq data. By comparison, though GLUE achieves higher label transfer accuracy than scBridge on the original data, its performance becomes worse and unstable when the data is contaminated with dropout noises. Similarly, scJoint achieves a comparable silhouette score with scBridge, but it encounters prominent performance reduction as the dropout rate increases. Fig. 5a–b and Supplementary Fig. 8 demonstrate the superiority of scBridge over six baselines in terms of data integration and label transfer. Such robustness of scBridge could attribute to its iterative and heterogeneous integration paradigm. Namely, even if the transcriptome is of low capture rate, scBridge could still identify a portion of reliable scATAC-seq data, which further helps the model to integrate the rest cells. Likewise, scBridge also achieves superior integration performance when the scATAC-seq data is contaminated with various degrees of dropout corruption. Note that some results do not exactly match those reported in the scJoint paper⁴ due to the differences in data preprocessing and the added dropout corruption.

[Point #10] Line 316-317, I noticed you applied the TF-IDF to the gene activity matrix. Do you have any references to do so? How do you scale the data to follow normal distribution?

[Response #10] We noted that Seurat⁵ preprocesses the scATAC-seq peak data with TF-IDF transformation, which both normalizes across cells to correct for differences in cellular sequencing depth, and across peaks to give higher values to more rare peaks. Motivated by its success, we tried the TF-IDF preprocessing for the scATAC-seq gene activity matrix and found it produces better results than the RNA-like preprocessing (*i.e.*, multiplied to have 10,000 counts per cell, log normalized, and scaled to have unit variance and zero means). Thus, we adopted TF-IDF to preprocess the gene activity matrix in our implementation. To provide a comprehensive understanding of the influence of different preprocessing strategies, we reported the performance of scBridge with TF-IDF and RNA-like preprocessing for scATAC-seq data on all seven datasets. The comparisons show that no matter which preprocessing strategy is adopted, scBridge still outperforms the most competitive baseline scJoint. The results were supplied in the Supplementary Note 3, which we attached below for your convenience.

Supplementary Note 3. Different preprocessing strategies for scATAC-seq data.

We noted that Seurat⁵ preprocesses the scATAC-seq peak data with TF-IDF transformation, which both normalizes across cells to correct for differences in cellular sequencing depth, and across peaks to give higher values to more rare peaks. Motivated by its success, we tried the TF-IDF preprocessing for the gene activity matrix and found it produces promising results as well.

To provide a comprehensive investigation on the TF-IDF preprocessing, we evaluated scBridge with TF-IDF and RNA-like preprocessing for scATAC-seq data. According to the five quantitative metrics on seven datasets shown in Fig. 10, scBridge with TF-IDF preprocessing achieves better performance in most cases than with RNA-like preprocessing. In addition, no matter which preprocessing strategy is adopted, scBridge still outperforms the most competitive baseline scJoint.

For the second question, we apologize for the typo of “scaling the data to follow normal distribution”. What we would like to express is scaling the scATAC-seq data to have unit variance and zero means just like we applied to scRNA-seq data. The scaling is achieved by the scale function provided in the Scanpy Python package¹⁸, v1.9.1. We corrected the words in the revision as follows.

Data preprocessing.

scBridge accepts the gene expression matrix of scRNA-seq data and the gene activity matrix of scATAC-seq data as the inputs. For scRNA-seq data, we first normalized each cell by dividing its total number of read counts on all genes, and then multiplied them by 10,000 to ensure that total counts are the same across cells. After that, we log normalized the read counts and scaled the data to have unit variance and zero means. For scATAC-seq data, we first applied the TF-IDF transformation on the gene activity matrix (more discussions are provided in Supplementary Note 3), and then scaled the data to have unit variance and zero means as well.

Supplementary Figure 10. Quantitative results of scBridge with different scATAC-seq data preprocessing strategies, compared with the most competitive baseline scJoint. Each boxplot ranges from the upper and lower quartiles with the median as the horizontal line and whiskers extend to 1.5 times the interquartile range. In the figure, “scBridge w/o TFIDF on scATAC-seq data” denotes that the gene activity matrix is preprocessed like scRNA-seq data, namely, multiplied to have 10,000 counts per cell, log normalized, and scaled to have unit variance and zero means.

Point-by-point Response to Reviewer #2

[Overall Comment] In this manuscript, Li and colleagues proposed an algorithm, scBridge, for integrating scRNA-seq data and scATAC-seq data. The proposed method accounts for varying cross-omics differences in different cell populations. The authors applied scBridge to five datasets, which showed the reliability of their proposed method.

[Response] Thanks for your positive feedback.

[Point #1] Would that be possible to benchmark with GLUE and Portal, which are also deep learning-based methods that can integrate scRNA-seq and scATAC-seq datasets?

- Cao, Z. J., & Gao, G. (2022). Multi-omics single-cell data integration and regulatory inference with graph-linked embedding. *Nature Biotechnology*, 40(10), 1458-1466.
- Zhao, J., Wang, G., Ming, J., Lin, Z., Wang, Y., Wu, A. R., & Yang, C. (2022). Adversarial domain translation networks for integrating large-scale atlas-level single-cell datasets. *Nature Computational Science*, 2(5), 317-330.

[Response #1] According to your comments, in addition to scJoint⁴, Seurat⁵, Harmony⁶, Conos⁷, we further included GLUE⁸ and Portal⁹ as baselines for comparison. In the revision, scBridge and six baselines are evaluated on seven multi-omics datasets (including two newly added golden benchmarks SHARE-seq and 10x Multiome). Notably, as Portal itself does not support label transfer, we adopted the KNeighborsClassifier to transfer cell annotations from scRNA-seq to scATAC-seq data, consistent with what we did for Harmony.

In the following, we attached the revised section of *Baseline methods*. The performance of the two baselines is added to Fig. 2–6 and Supplementary Fig. 2–9 in the revision. In short, scBridge achieves superior performance on seven datasets in terms of data integration and label transfer compared with six baseline methods.

Performance and benchmarking.

Baseline methods.

Six single-cell data integration methods were benchmarked for comparisons, including scJoint⁴, Seurat⁵, Portal⁹, Harmony⁶, GLUE⁸, and Conos⁷.

For scJoint⁴, we adopted the official code released in <https://github.com/SydneyBioX/scJoint>, with hyper-parameters provided in the code example or recommended in the paper. Since scJoint first binarizes the gene expression and activity matrices, no data preprocessing is needed. The algorithm directly outputs the embedding of both scRNA-seq and scATAC-seq data (*_embeddings.txt*), the transferred label for scATAC-seq data (*_knn_predictions.txt*), as well as the confidence score (*_knn_probs.txt*).

For Seurat, we used Seurat R package⁵, v4.1.4. The raw count matrix of scRNA-seq and the unnormalized gene activity matrix of scATAC-seq were transformed into Seurat objects using the CreateSeuratObject function. With the NormalizeData, FindVariableFeatures, ScaleData, and RunPCA functions, the Seurat objects were further processed, where the PCA dimensions were set to 1 : 30 for scRNA-seq data and 2 : 30 for scATAC-seq data, respectively. After that, the FindTransferAnchors function was adopted to identify the anchors between scRNA-seq and scATAC-seq datasets. Then, the TransferData function was employed to impute and integrate the scATAC-seq data, and transfer the annotations from the scRNA-seq dataset into the scATAC-seq cells. The TransferData function also outputs the confidence score of each prediction. Finally, PCA was performed on the combined matrix of scRNA-seq data and imputed scATAC-seq data to obtain features. If not mentioned, all parameters are set as default.

For Portal, we used the portal-sc Python package⁹, v1.0.2. Following its default pipeline, we used the model.preprocess function to preprocess the gene expression matrix and gene activity matrix. After that, the model.train function was used to integrate data. We set *training_steps* = 1,000 for datasets with sample size < 20,000 and *training_steps* = 2,000 otherwise, as suggested in the package. Finally, the integration results were obtained from model.latent after running the model.eval function. As Portal itself does not support label transfer, we adopted the KNeighborsClassifier function with *k* = 10 provided in the scikit-learn Python package¹⁴, v1.1.3, to transfer cell annotations from scRNA-seq to scATAC-seq data.

For Harmony, we adopted the same data preprocessing pipeline as scBridge, followed by a PCA dimensional reduction with 50 components, and then used the official harmony Python package⁶ (<https://github.com/slowkow/harmony>), v0.0.5, with the recommended parameters for data integration. As Harmony itself does not support label

transfer, we adopted the KNeighborsClassifier function with $k = 10$ provided in the scikit-learn Python package¹⁴, v1.1.3, to transfer cell annotations from scRNA-seq to scATAC-seq data.

For GLUE, we adopted the official code released in <https://github.com/gao-lab/GLUE>. Following its default pipeline, we first preprocessed the scRNA-seq expression and scATAC-seq peak-by-cell matrices and then constructed the prior regulatory graph. After that, we trained a glue model to integrate data with the `scglue.models.fit_SCGLUE` function. Then, we applied the model for cell and feature embedding using the `encode_data` function. Finally, cell annotations with confidence scores were transferred from scRNA-seq to scATAC-seq data with the `scglue.data.transfer_labels` function.

For Conos, we used conos R package⁷, v1.4.6. The raw count matrix of scRNA-seq and the unnormalized gene activity matrix of scATAC-seq were first preprocessed by the `basicP2proc` function provided in pagoda2 R package¹⁵, v1.0.10. Then, the joint graph was built by the `bulidGraph` function with parameters $k = 15$, $k.self = 5$, $k.self.weigh = 0.01$, $ncomps = 30$, $n.odgenes = 5e3$, and $space = 'PCA'$. The joint embedding was generated by the `embedGraph` function. Finally, labels were propagated from the scRNA-seq to the scATAC-seq based on the joint graph using the `propagateLabels` function, which also outputs the confidence score of each prediction. All parameters are set as default if not mentioned.

[Point #2] The authors mentioned that there is an optional loss in scBridge named structure preservation loss, which can handle the case that there are some scATAC-seq-unique cell types. They applied this loss on the integration of human myocardial infarction data. But in general, without scATAC-seq annotations, it is hard to anticipate such a situation. The authors may need to provide users with more clear guidance about when to use the structure preservation loss.

[Response #2] Thanks for the practical concern and valuable suggestion. As suggested, we proposed a guidance on when to enable the structure preservation loss, namely, through observing the pattern differences of unconfident cells with and without the structure preservation loss. To be specific, the impact of the structure preservation loss is different for cells of seen and novel types due to the following reasons. Notably, the structure preservation loss intrinsically preserves dissimilarity of scATAC-seq cells from the raw space into the embedding space. Hence, for the scATAC-seq cells of seen types, the structure preservation loss would have little influence on them since within-class cells are similar in the raw space. As a result, they would be correctly integrated with scRNA-seq cells. In contrast, for cells of novel types, the structure preservation loss would prevent them from merging into clusters of seen types since they are usually dissimilar in the raw space.

Consequently, as illustrated in Supplementary Fig. 6f, when there are indeed novel scATAC-seq cells, enabling the structure preservation loss would form a clear cluster of unconfident cells, compared with the UMAP visualization when the loss is disabled (see the left two figures). On the contrary, when there are no novel scATAC-seq cells, only slight differences would be observed between the UMAP visualization with and without the structure preservation loss. Namely, unconfident cells are commonly positioned at the boundary of clusters (see the right two figures). In summary, users could decide whether to enable novel type discovery by observing the pattern differences of unconfident cells with and without the structure preservation loss.

We provided detailed guidance in a new subsection in the Supplementary Note 2. For your convenience, we attached the subsection and the corresponding figures below.

Supplementary Figure 6. **a**, UMAP visualization of Portal and Harmony on the human myocardial infarction data with novel types. The first and second rows show cells colored by types and omics, respectively. **b**, UMAP visualization of scATAC-seq cells obtained by scJoint, Seurat, GLUE, and Conos, where cells are colored by the confidence score. **c**, The label transfer results of scJoint. Cells with a confidence score below the GMM-estimated threshold are considered novel. **d**, The confidence score for different types of scATAC-seq cells predicted by scJoint, GLUE, and Conos. **e**, The novel type threshold of scJoint, Seurat, GLUE, and Conos, which is estimated by applying a two-component GMM on the confidence score. **f**, UMAP visualization of scATAC-seq cells obtained by scBridge, where cells colored by the confidence score on four cases. Namely, from left to right, enabling structure loss on data with novel cell type, disabling structure loss on data with novel cell type, enabling structure loss on data without novel cell type, and disabling structure loss on data without novel cell type.

Supplementary Note 2. How and when to discover novel types?

Here we supplied the detailed pipeline of novel type discovery. After the training with the structure preservation loss L_{STC} , scBridge would compute a confidence score ranged $[0, 1]$ for each of the n^t scATAC-seq cells, denoted as $r^t = \{r_1^t, r_2^t, \dots, r_{n^t}^t\}$. To predict which portion of cells is novel, we derived a confidence threshold in a data-driven manner. Specifically, we fit the distribution of r^t with a two-component GMM g , namely,

$$g(r^t) = \gamma_{c_1} \phi(d^t | c_1) + \gamma_{c_2} \phi(d^t | c_2), \quad (17)$$

where $\gamma_{c_1}, \gamma_{c_2}$ denote the mixture coefficient for components c_1, c_2 . Let $p(c_1 | r_i^t), p(c_2 | r_i^t) \in [0, 1]$ be the probability of cell i belonging to the two components, the confidence threshold dividing novel and seen cell types is estimated as β that satisfies $p(c_1 | r_i^t) = p(c_2 | r_i^t)$. If there is more than one suitable $\beta \in [0, 1]$, we chose the smallest one as the confidence threshold. Finally, those scATAC-seq cells with $r_i^t < \beta$ are predicted as novel. The confidence score distribution, the two GMM components, and the estimated confidence threshold of scBridge and other baselines are illustrated in Fig. 4b and Supplementary Fig. 6e.

Next, to handle the occasion where users do not have enough prior information to decide whether there are novel cell types in scATAC-seq data, we provided guidance on when to enable the structure preservation loss of scBridge for novel type discovery. Specifically, we encourage the users to run scBridge twice with and without the structure preservation loss respectively, and visualize the scATAC-seq cell embedding colored by the confidence score. As illustrated in Supplementary Fig. 6f, when there are indeed novel scATAC-seq cells, enabling the structure preservation loss would form a clear cluster of unconfident cells, compared with the UMAP visualization when the loss is disabled (see the left two figures). On the contrary, when there are no novel scATAC-seq cells, only slight differences would be observed between the UMAP visualization with and without the structure preservation loss. Namely, unconfident cells are commonly positioned at the boundary of clusters (see the right two figures). In summary, users could decide whether to enable novel type discovery by observing the pattern differences of unconfident cells with and without the structure preservation loss. Notably, the structure preservation loss intrinsically enables scBridge to discover a portion of cells that are most different from annotated scRNA-seq cells, which is practically useful aided with additional manual analysis on those cells.

[Point #3] The authors may need to compare the computation efficiency of scBridge with other methods.

[Response #3] Thanks for the suggestion. In the revision, we evaluated the running time and memory consumption of scBridge on mouse atlas subsets of 5,000–80,000 cells compared with six baselines. The results in Fig. 2c show that scBridge requires linearly increasing time with respect to cell numbers, and achieves the third-best time efficiency. Moreover, scBridge requires a constant memory consumption for the arbitrary number of cells and achieves the second-best memory efficiency. The linear time complexity and constant memory complexity make scBridge a promising tool for handling large-scale data.

For your convenience, we attached the corresponding revisions below.

scBridge scales to atlas data.

With the development of sequencing techniques, the number of cells profiled with various protocols grows continually, arousing the demand for efficiently handling large-scale data. To access how scBridge scales to large data, we evaluated it on the mouse atlas dataset. Specifically, we used the cells sequenced with FACS and droplet protocols provided by Tabula Muris¹² as scRNA-seq data, and the cells sequenced by Cusanovich et al.¹³ as scATAC-seq data. After data preprocessing, 102,103 cells from 18 common types are selected for evaluation.

To investigate the computation efficiency of scBridge, we applied it to five subsets of mouse atlas with 5,000–80,000 cells. Fig. 3c shows the (logged) running time and memory consumption of all tested methods with respect to different cell numbers. As shown, scBridge takes linearly increasing running time (the third-best) and constant memory consumption (the second-best), which is favorable in scaling to large data.

Figure 3. Integration results on mouse atlas data. **a**, UMAP visualization of the joint embeddings learned by scBridge and scJoint. The first and second rows show cells colored by types and omics, respectively. **b**, The cell type and 1–omics silhouette coefficients of scBridge and six baselines. A higher cell type silhouette coefficient indicates better biological grouping, and a higher 1–omics silhouette coefficient indicates better cross-omics cell mixing. Each boxplot ranges from the upper and lower quartiles with the median as the horizontal line and whiskers extend to 1.5 times the interquartile range. **c**, The running time and memory consumption of different methods on mouse atlas subsets of 5,000–80,000 cells. **d**, The dot plot of relative expression of marker genes *Tgfb1*, *Cd68*, *Cd34* in cells predicted as Hematopoietic stem cell (HSC), Macrophages, and Monocytes by scBridge. The size of the circle represents the proportion of expressing cells, and the color indicates the average expression level. **e**, The agreement between the predicted label and the manual annotation. Matrices with a clearer diagonal structure indicate better performance.

References

1. Chen, S., Lake, B. B. & Zhang, K. High-throughput sequencing of the transcriptome and chromatin accessibility in the same cell. *Nat. biotechnology* **37**, 1452–1457 (2019).
2. Ma, S. *et al.* Chromatin potential identified by shared single-cell profiling of rna and chromatin. *Cell* **183**, 1103–1116 (2020).
3. Mouse kidney nuclei isolated with chromium nuclei isolation kit, single cell multiome atac + gene expression dataset by cell ranger arc 2.0.2. *10x Genomics* (2023). <https://www.10xgenomics.com/resources/datasets/mouse-kidney-nuclei-isolated-with-chromium-nuclei-isolation-kit-saltyez-protocol-and-10x-complex-tissue-dp-ct-sorted-and-ct-unsorted-1-standard>.
4. Lin, Y. *et al.* scjoint integrates atlas-scale single-cell rna-seq and atac-seq data with transfer learning. *Nat. Biotechnol.* **40**, 703–710 (2022).
5. Stuart, T. *et al.* Comprehensive integration of single-cell data. *Cell* **177**, 1888–1902 (2019).
6. Korsunsky, I. *et al.* Fast, sensitive and accurate integration of single-cell data with harmony. *Nat. methods* **16**, 1289–1296 (2019).
7. Barkas, N. *et al.* Joint analysis of heterogeneous single-cell rna-seq dataset collections. *Nat. methods* **16**, 695–698 (2019).
8. Cao, Z.-J. & Gao, G. Multi-omics single-cell data integration and regulatory inference with graph-linked embedding. *Nat. Biotechnol.* **40**, 1458–1466 (2022).
9. Zhao, J. *et al.* Adversarial domain translation networks for integrating large-scale atlas-level single-cell datasets. *Nat. Comput. Sci.* **2**, 317–330 (2022).
10. Muto, Y. *et al.* Single cell transcriptional and chromatin accessibility profiling redefine cellular heterogeneity in the adult human kidney. *Nat. communications* **12**, 2190 (2021).
11. Carter, B. & Zhao, K. The epigenetic basis of cellular heterogeneity. *Nat. Rev. Genet.* **22**, 235–250 (2021).
12. Consortium, T. M. *et al.* Single-cell transcriptomics of 20 mouse organs creates a tabula muris. *Nature* **562**, 367–372 (2018).
13. Cusanovich, D. A. *et al.* A single-cell atlas of in vivo mammalian chromatin accessibility. *Cell* **174**, 1309–1324 (2018).
14. Pedregosa, F. *et al.* Scikit-learn: Machine learning in Python. *J. Mach. Learn. Res.* **12**, 2825–2830 (2011).
15. Barkas, N., Petukhov, V., Kharchenko, P. & Biederstedt, E. pagoda2: single cell analysis and differential expression. *R package version* **102** (2021).
16. Granja, J. M. *et al.* Single-cell multiomic analysis identifies regulatory programs in mixed-phenotype acute leukemia. *Nat. biotechnology* **37**, 1458–1465 (2019).
17. McCarthy, D. J., Campbell, K. R., Lun, A. T. & Wills, Q. F. Scater: pre-processing, quality control, normalization and visualization of single-cell rna-seq data in r. *Bioinformatics* **33**, 1179–1186 (2017).
18. Wolf, F. A., Angerer, P. & Theis, F. J. Scanpy: large-scale single-cell gene expression data analysis. *Genome biology* **19**, 1–5 (2018).

REVIEWER COMMENTS

Reviewer #1 (Remarks to the Author):

Most of my comments have been addressed except for Point #9. The peak-by-cell matrix can also be downsampled using the same strategy as a gene-by-cell matrix. This shouldn't be the excuse to exclude GLUE.

Reviewer #2 (Remarks to the Author):

Thank all the authors' efforts to address my comments. I don't have other concerns.

Point-by-point Response to Reviewer #1

[Comment] Most of my comments have been addressed except for Point #9. The peak-by-cell matrix can also be downsampled using the same strategy as a gene-by-cell matrix. This shouldn't be the excuse to exclude GLUE.

[Response] Thanks for your positive feedback and insightful comment. As pointed out, the dropout events could be approximated by downsampling the peak-by-cell matrix using the same strategy as a gene-by-cell matrix, especially under multiple random downsampling experiments. According to your advice, we evaluated GLUE on scATAC-seq dropout data in Fig. 5c, which shows that scBridge outperforms GLUE in most cases.

For your convenience, we attached the revised figure and manuscript below.

Figure 5. Integration results on human hematopoiesis data. **a**, UMAP visualization of the joint embeddings learned by scBridge, scJoint, Seurat, and Portal under 75% dropout on scRNA-seq data, where cells are colored by types. **b**, The agreement between labels transferred by scBridge, Seurat, and the manual annotations under 75% dropout on scRNA-seq data. A clearer diagonal structure indicates better agreement. **c**, The F1 harmonized silhouette score and the weighted F1 label transfer accuracy of scBridge and six baselines with different dropout corruption rates on scRNA-seq and scATAC-seq data. Five random experiments are conducted under each dropout rate. Each boxplot ranges from the upper and lower quartiles with the median as the horizontal line and whiskers extend to 1.5 times the interquartile range.

To investigate the robustness of scBridge against the dropout technical noise, we applied it to the human hematopoiesis data which contains 34,609 scRNA-seq and 33,819 scATAC-seq cells from 23 common types. To simulate the dropout events, we downsampled the scRNA-seq count matrix, scATAC-seq activity matrix, and scATAC-seq peak-by-cell matrix by 25%, 50%, and 75%, respectively, with the `downsampleMatrix` function provided in the `scuttle` R package¹. As shown in Fig. 5c, scBridge achieves superior robustness towards the scRNA-seq data quality. Namely, its integration and label transfer performances are almost impervious under up to 75% dropout rate on scRNA-seq data. By comparison, though GLUE achieves higher label transfer accuracy than scBridge on the original data, its performance becomes worse and unstable on data contaminated with dropout noises. Similarly, scJoint achieves a comparable silhouette score with scBridge, but encounters prominent performance reduction as the dropout rate increases. Fig. 5a–b and Supplementary Fig. 8 demonstrate the superiority of scBridge over six baselines in data integration and label transfer. Likewise, scBridge also achieves better performance on the corrupted scATAC-seq data, especially under high dropout rates as shown in Fig. 5c. Such robustness of scBridge could attribute to its iterative and heterogeneous integration paradigm. Namely, even if the sequencing data is of low capture rate, scBridge could still identify a portion of reliable scATAC-seq data, which further helps the model to integrate the rest cells. Note that some results do not exactly match those reported in the scJoint paper² due to the differences in data preprocessing and the added dropout corruption.

Point-by-point Response to Reviewer #2

[Comment] Thank all the authors' efforts to address my comments. I don't have other concerns.

[Response] We sincerely appreciate your kind comments. Thank you.

References

1. McCarthy, D. J., Campbell, K. R., Lun, A. T. & Wills, Q. F. Scater: pre-processing, quality control, normalization and visualization of single-cell rna-seq data in r. *Bioinformatics* **33**, 1179–1186 (2017).
2. Lin, Y. et al. scjoint integrates atlas-scale single-cell rna-seq and atac-seq data with transfer learning. *Nat. Biotechnol.* **40**, 703–710 (2022).

REVIEWERS' COMMENTS

Reviewer #1 (Remarks to the Author):

All my comments have been addressed.

Point-by-point Response to Reviewer #1

[Comment] All my comments have been addressed.

[Response] We sincerely appreciate your positive feedback and previous constructive comments. Thank you.